# Breaking the Lock-in:
# Diversifying Text-to-Image Generation via Representation Modulation

**Dahee Kwon** [1]  **Haeun Lee** [1] [†]  **Jaesik Choi** [1] [2]

## Abstract

Recent text-to-image models built on large-scale Transformer backbones and flow-based objectives deliver strong text–image alignment and high visual quality, yet often produce overly similar samples under a fixed prompt. Existing diversity-enhancement methods alleviate this, but typically require expensive sampling or auxiliary optimization, incurring non-trivial overhead. To investigate the root cause of this homogeneity, we examine intermediate Transformer features and observe that the zero-frequency spatial average (DC) component rapidly converges across seeds early in generation, causing early trajectory lock-in that limits downstream variation. Building on this, we propose DC Attenuation for diVersity Enhancement (**DAVE**), a training-free representation-level intervention that selectively attenuates this component in the early regime. DAVE preserves the sampling pipeline with negligible overhead, improving prompt-consistent diversity while maintaining competitive image quality.

## 1. Introduction

Recent advancements in text-to-image (T2I) generative models—particularly those leveraging scalable Transformer architectures and flow-based objectives—have established these models as a dominant paradigm in generative modeling, featuring remarkable text–image alignment and photorealism. These models enable users to reliably produce high-quality visual content, driving widespread adoption across diverse domains (Croitoru et al., 2023; Esser et al., 2024; Chen et al., 2025).

However, this reliability in quality is often accompanied by a reduction in generation diversity, defined as variation in under-specified visual factors (e.g., layout, style) while preserving prompt-conditioned semantics. When generating multiple images from a fixed prompt, outputs often exhibit limited variation, converging to similar compositions or stylistic patterns (Mukhopadhyay et al., 2023; Astolfi et al., 2024; Albuquerque et al., 2025). Insufficient diversity hampers users' ability to explore broad candidates, discover rare configurations, and construct synthetic datasets with wide distributional coverage, ultimately reducing downstream performance gains. Consequently, diversity is not merely an optional add-on, but a core performance axis that determines the usability and scalability of generative models.

A growing body of research has recently turned its attention to this issue, with many approaches introducing diversity control directly at sampling time. By tuning guidance schedules (Um & Ye, 2025; Sadat et al., 2023), injecting stochastic perturbations (Harrington et al., 2025), or modifying the sampling process (Morshed & Boddeti, 2025; Corso et al., 2023), these training-free methods can steer generation behavior to recover diversity while preserving visual quality.

However, existing approaches face two practical limitations. First, they often incur substantial computational and memory overhead. Enhancing diversity typically requires extra sampling steps, auxiliary optimization, or parallel candidate generation across multiple seeds—all of which exacerbate memory consumption and decoding costs. As modern generative models scale, even modest overheads become a significant bottleneck. Second, the fundamental causes of limited diversity remain insufficiently understood. While prior studies analyze diversity through inference-time controls, a detailed mechanistic account of exactly where and why this collapse occurs within the model's internal representations remains largely unexplored.

From this perspective, we approach diversity control through the lens of representation-level interventions. By analyzing intermediate Transformer representations, we observe that the zero-frequency spatial average (DC) compo-

---

[†]Work done during an undergraduate internship at KAIST. [1]Korea Advanced Institute of Science and Technology (KAIST) [2]INEEJI. Correspondence to: Jaesik Choi <jaesik.choi@kaist.ac.kr>.

*Proceedings of the 43rd International Conference on Machine Learning*, Seoul, South Korea. PMLR 306, 2026. Copyright 2026 by the author(s).

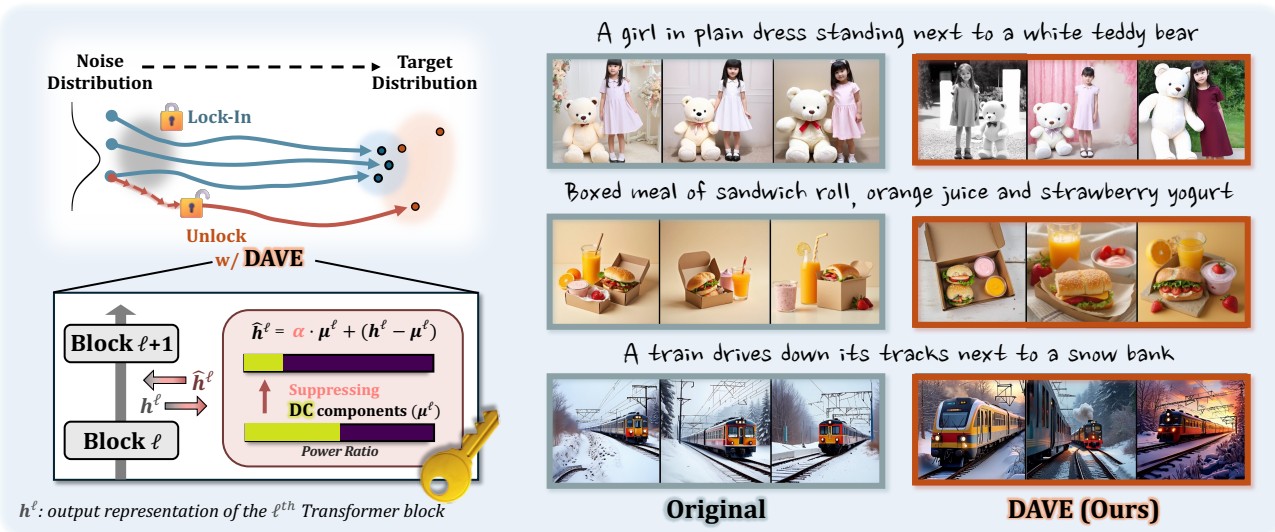

*Figure 1.* Overview of DAVE. Compared to the original flow-based generative model, DAVE consistently produces more diverse images with minimal computation.

nent exhibits strong trajectory lock-in in the early denoising steps across seeds—a phase that coincides with the formation of global layouts and coarse semantic structures. Motivated by this observation, we propose DC Attenuation for diVersity Enhancement (DAVE), which selectively attenuates this component at inference time via a lightweight internal operation. Requiring neither retraining nor changes to the sampling pipeline, this simple adjustment incurs virtually no computational or memory overhead and imposes no batch size constraints. Through extensive experiments, we demonstrate that DAVE achieves competitive performance against a range of state-of-the-art methods while incurring substantially lower overhead.

## 2. Related Work

**Diversity Enhancement** Recent efforts to mitigate diversity collapse in T2I models broadly fall into two categories: batch-wise diversification and individual trajectory manipulation. Most approaches adopt the former, explicitly promoting diversity within a batch by encouraging jointly generated samples to diverge. For instance, Particle Guidance (PG) (Corso et al., 2023) adds a pairwise gradient potential to push samples apart, while DiverseFlow (Morshed & Boddeti, 2025) optimizes a batch-wide diversity objective along the trajectory. To refine this repulsion, SPELL (Kirchhof et al., 2024) introduces sparse repellency terms that activate only when trajectories risk collapsing onto one another. Similarly, OSCAR (Wu et al., 2025) maximizes the feature-space volume of a batch and injects stochasticity projected orthogonally to the flow, spreading trajectories

without harming fidelity. Extending this concept beyond a single batch, SPARKE (Jalali et al., 2026) guides sampling with a prompt-aware Rényi kernel entropy score to scale efficiently across large prompt sets. However, these joint-evaluation methods incur non-trivial computational and memory overheads, which become increasingly burdensome as generative models scale up.

An alternative line of approach manipulates each trajectory directly rather than comparing samples. For example, CADS (Sadat et al., 2023) injects scheduled noise into the text condition during early inference to prevent over-reliance on the prompt and recover diversity suppressed at high guidance scales. Yet, these methods still operate primarily as sampling-level heuristics. They offer little insight into the internal mechanisms that drive generations toward similar outputs in the first place, leaving the root cause of diversity collapse largely unexamined.

**Internal Representation Analysis** A complementary line of research examines the internal representations of T2I models to improve their interpretability and controllability (Kim et al., 2025; Shin et al., 2025; Li et al., 2026; Yang et al., 2025; Dalva et al., 2024; Si et al., 2024; Han et al., 2025). Several works manipulate intermediate features to steer diffusion or flow dynamics at inference time, such as attention-based editing (Cao et al., 2023; Voynov et al., 2023; Hertz et al., 2022) and architectural rebalancing (Si et al., 2024). Beyond controllable editing, modulating internal features can also foster generation creativity, as seen in C3 (Han et al., 2025). Yet, even such diversity-oriented approaches overlook the fundamental mechanism of diver-

sity collapse. They rarely investigate where and when the trajectory locks in. In contrast, we analyze the internal feature dynamics of Transformer backbones, pinpointing early DC convergence as a major representational bottleneck for seed-level variation. Consequently, we propose DAVE, a direct representation modulation approach that enhances diversity without additional sampling constraints.

## 3. Method

In this section, we first investigate the representation-level mechanisms driving diversity collapse, motivating our proposed method: DC Attenuation for diVersity Enhancement (DAVE). By selectively attenuating the internal DC component during the early stages of generation, DAVE broadens the model's generative scope with minimal complexity.

### 3.1. Preliminary

Text-conditioned flow matching generates samples by transporting a source distribution $X_0$ to the data distribution $X_1$. In practice, we take $p_0 = \mathcal{N}(0, I)$, sample $x_0 \sim p_0$, and evolve it over time via a learned vector field. The resulting trajectory $x_t$ follows the Ordinary Differential Equation:

$$\frac{dx_t}{dt} = v_\theta(x_t, t; c), \qquad t \in [0, 1], \tag{1}$$

where $c$ denotes the conditioning signal (e.g., a text embedding) and $v_\theta(x_t, t; c)$ specifies the instantaneous direction of evolution (velocity) at time $t$.

Sampling is performed by discretizing Eq. (1) into $K$ steps. Starting from $x_0 \sim p_0$, we iteratively update the state according to a numerical solver:

$$x_{k+1} = x_k + \Delta t \, v_\theta(x_k, t_k; c), \qquad k = 0, \ldots, K-1, \tag{2}$$

where $t_k$ and $\Delta t$ denote the discretized time and step size, respectively. The final generated sample is $x_K \approx x_1$.

In modern text-to-image architectures, the vector field $v_\theta$ generally adopts a Transformer backbone. Let $h_t^{(\ell)}$ denote the hidden representation at block $\ell$. A single Transformer block updates the representation as:

$$h_t^{(\ell+1)} = \text{Block}^{(\ell+1)}(h_t^{(\ell)}, t, c). \tag{3}$$

The velocity prediction $v_\theta(x_t, t; c)$ at step $t$ is computed from the last-block representation.

### 3.2. Lock-in on DC component

To investigate the mechanisms underlying this diversity degradation, we analyze the internal representations of transformer blocks. This investigation reveals a notable pattern:

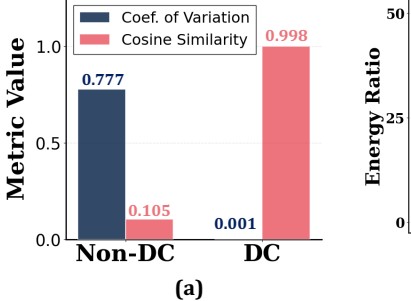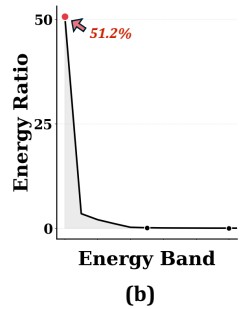

*Figure 2.* (left) DC variation across seeds; "non-DC" denotes representations with the DC component removed. (right) Band energy ratios (band/total), where the leftmost bin corresponds to the DC (lowest-frequency) component.

a pervasive global bias that manifests as a systematic drift in hidden states during generation. Specifically, a comparison of hidden representations $h_t^{(\ell)}$ across various noise seeds shows that the zero-frequency component—namely, the DC component—becomes remarkably aligned across samples. Using SD3 (Esser et al., 2024), we analyze representations from Transformer block 5 over 100 random seeds per prompt. The DC component shows high consistency across these seeds, exhibiting high pairwise cosine similarity and a low coefficient of variation (Figure 2). Accounting for 51.2% of the total energy, this component constitutes a dominant global signal rather than a minor artifact—a phenomenon we term *early DC drift*.

We posit that early DC drift is a major bottleneck of reduced sample diversity, arising from a synergy between architectural bias and the training objective. Since the DC component represents the zero-frequency spatial average, its early dominance is directly explained by the spectral bias of neural networks (Rahaman et al., 2019; Wang & Pehlevan, 2026)—their tendency to prioritize low-frequency signals over high-frequency details. This bias directs the model's focus toward the DC component at the onset of sampling, consistent with the early emergence of global structures in the generative process (Choi et al., 2021; Liu et al., 2025; Esser et al., 2024).

This tendency is further exacerbated by MSE-based flow-matching objectives under high uncertainty. In early sampling steps characterized by a low Signal-to-Noise Ratio (SNR), the objective is mathematically minimized when the model predicts the text-conditioned expectation of the data. While high-frequency, sample-specific textures average out across seeds, the DC component remains a robust cue of global statistics. Driven by this dual bias, the DC component is strongly pulled toward the conditional mean, establishing a seed-invariant anchor that dictates the generative trajectory. Consequently, the global layout tends to

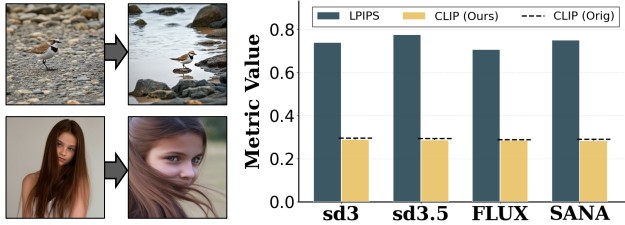

*Figure 3.* (Left) Outputs with DC attenuation for "A small bird standing on a rocky beach." and "A beautiful girl with long brown hair." (Right) Perceptual changes and text–image alignment under DC attenuation; CLIP (Orig) is the unmanipulated CLIP score.

lock in so early that it severely restricts the ability of the initial noise seed to induce sufficient structural variation. A step-wise analysis of the denoising trajectory confirms that this cross-seed DC alignment is indeed concentrated in the earliest steps and decays toward the end of generation (See Table 3). Appendix E formalizes how early DC-dominated alignment can limit trajectory separation and diversity.

### 3.3. DC Attenuation for Diversity Enhancement

Given that early DC drift acts as a restrictive global anchor, we hypothesize that directly weakening its influence can unlock the generative trajectory from its premature structural commitment. To test this, we selectively attenuate the DC component during early sampling steps to suppress this dominant global bias. As shown in Figure 3, this targeted intervention substantially changes the final image layouts without compromising semantic alignment. These observations confirm that modulating the early-stage DC component is a highly effective strategy for diversity enhancement.

Building on these empirical insights, we propose DC Attenuation for diVersity Enhancement (DAVE). DAVE is designed to broaden the model's generative scope by strategically dampening the dominant DC signal during early generation stages. By dismantling this seed-invariant anchor, the method empowers the stochasticity of the initial noise to drive meaningful structural variations, breaking the homogenizing effect of the conditional mean.

For a hidden representation $h_t^{(\ell)} \in \mathbb{R}^{D \times C}$, where $D$ is the number of visual tokens and $C$ is the channel dimension, the DC component $\mu_t^{(\ell)}$ is straightforwardly obtained as the spatial mean, bypassing the need for explicit frequency-domain transforms:

$$\mu_t^{(\ell)} = \frac{1}{D} \sum_{d=1}^{D} h_t^{(\ell)}[d, :] \ \in \ \mathbb{R}^{1 \times C}. \tag{4}$$

Since our focus is on spatial structures, we isolate and manipulate only the representations containing visual informa-

tion, particularly in multi-modal architectures like MMDiT. DAVE intervenes on the output of Transformer blocks $\ell \in \mathcal{L}$ during the early generative phase ($t < \tau$):

$$\hat{h}_t^{(\ell)} = \alpha \cdot \mu_t^{(\ell)} + \left( h_t^{(\ell)} - \mu_t^{(\ell)} \right). \tag{5}$$

where $\alpha \in (0, 1)$ controls the attenuation strength, and $\mu_t^{(\ell)}$ is broadcast across all D tokens when subtracted from (and added to) $h_t^{(\ell)}$. We substitute the original hidden representation $h_t^{(\ell)}$ with the modified $\hat{h}_t^{(\ell)}$ before it is passed to the subsequent Transformer block ($\ell + 1$). As a surgical, training-free intervention, DAVE integrates seamlessly into existing pre-trained models without requiring any architectural modifications or additional optimizations. By attenuating the early-stage DC component, DAVE prevents premature anchoring to a common structural baseline, thereby amplifying the relative influence of seed-specific spatial residuals.

**Parameter Selection.** DAVE is modulated by three key parameters: the attenuation strength $\alpha$, the target block pool $\mathcal{L}$, and the temporal cutoff $\tau$. We outline practical guidance for selecting each parameter below, with a comprehensive empirical analysis of these configurations in Section 4.2.

Temporal Cutoff ($\tau$): Following the principle of early-stage intervention, we apply DC attenuation during the early generative regime, i.e., for timesteps satisfying $t < \tau$. Based on the trade-off between output variation and perceptual quality observed in our ablations, we find that intervening within the first 15–20% of the generative process yields stable diversity gains across the evaluated architectures.

Attenuation Strength ($\alpha$): The coefficient $\alpha \in (0, 1)$ regulates the magnitude of DC suppression. While a smaller $\alpha$ induces more pronounced structural deviations, excessive attenuation may lead to perceptual artifacts or loss of semantic coherence. To balance structural diversification with text-alignment preservation, we recommend setting $\alpha \in [0.2, 0.5]$. This enables adjustable intervention intensity based on the desired level of variation.

Target block pool ($\mathcal{L}$): We define $\mathcal{L}$ as the set of Transformer blocks whose DC component shows strong cross-seed convergence, aligning across noise realizations. Empirically, DAVE is robust to block choice within this pool; interventions on blocks in $\mathcal{L}$ yield consistent diversity gains. Although the specific indices of $\mathcal{L}$ vary across architectures, these variations reflect structural differences rather than ad-hoc, per-model heuristics. Moreover, the identified blocks largely coincide with structurally significant modules reported in prior literature (Li et al., 2026; Yang et al., 2025), further supporting the architectural grounding of our selection process.

*Table 1.* Quantitative results for diverse generation in the independently sampled generation setting, where samples are generated without explicit in-batch interaction. Bold indicates the best result for each metric, and underlining indicates the second-best.

| Dataset | Model | Method | FID ↓ | Prec ↑ | Rec ↑ | Cov ↑ | Dens ↑ | Vendi ↑ | CLIP ↑ |
|---|---|---|---|---|---|---|---|---|---|
| **ImageNet** | **SD3.5** | Orig | 22.23 | 0.8604 | 0.2589 | 0.7757 | 1.0071 | 1.71 | 0.2952 |
| | | Orig ($\omega_{CFG}=2$) | **17.13** | 0.8144 | 0.5106 | 0.8032 | 0.9311 | 2.05 | 0.2861 |
| | | CADS | 17.91 | 0.7856 | 0.5698 | 0.7936 | 0.8386 | 2.09 | 0.2907 |
| | | SPARKE | 22.27 | **0.8686** | 0.3136 | 0.4004 | **1.2289** | 1.77 | **0.3016** |
| | | Ours | 20.74 | 0.8090 | **0.6489** | 0.7895 | 0.7810 | 2.33 | 0.2897 |
| | | Ours Random | 17.57 | 0.8591 | 0.5422 | **0.8371** | 1.0114 | **2.50** | 0.2939 |
| | **Flux.1-dev** | Orig | 24.01 | 0.8219 | 0.3149 | 0.7555 | 0.9371 | 1.88 | **0.2902** |
| | | Orig ($\omega_{CFG}=2$) | **21.00** | 0.8220 | 0.4064 | **0.7726** | 0.9475 | 2.09 | 0.2852 |
| | | CADS | 39.97 | 0.5914 | 0.5172 | 0.5913 | 0.5045 | **2.69** | 0.2689 |
| | | SPARKE | 22.65 | **0.8298** | 0.3561 | 0.3935 | **1.1112** | 2.04 | 0.2921 |
| | | Ours | 25.31 | 0.7112 | 0.5102 | 0.6963 | 0.7189 | 2.26 | 0.2858 |
| | | Ours Random | 24.36 | 0.6927 | **0.6032** | 0.7295 | 0.6869 | 2.44 | 0.2818 |
| | **SANA1.5** | Orig | 27.85 | 0.7860 | 0.1623 | 0.6620 | 0.8506 | 1.59 | 0.2918 |
| | | Orig ($\omega_{CFG}=2$) | **22.44** | 0.7986 | 0.2846 | **0.7182** | **0.8963** | 1.80 | 0.2827 |
| | | CADS | 68.13 | 0.3952 | **0.5774** | 0.4525 | 0.3062 | **2.32** | 0.2618 |
| | | SPARKE | 27.87 | 0.7774 | 0.2220 | 0.3013 | 0.9739 | 1.67 | **0.2935** |
| | | Ours | 25.16 | 0.7588 | 0.5254 | 0.6435 | 0.6422 | 2.20 | 0.2885 |
| | | Ours Random | 22.41 | 0.7419 | 0.4911 | 0.6990 | 0.7935 | 2.15 | 0.2879 |
| **MSCOCO** | **SD3.5** | Orig | 36.38 | **0.8208** | 0.2546 | 0.7928 | **1.1345** | 2.29 | **0.3129** |
| | | Orig ($\omega_{CFG}=2$) | **28.48** | 0.7864 | 0.4282 | 0.8222 | 0.9846 | 1.83 | 0.3047 |
| | | CADS | 28.97 | 0.7138 | 0.4604 | 0.7896 | 0.8232 | 2.30 | 0.3065 |
| | | SPARKE | 35.39 | 0.7938 | 0.2730 | 0.7792 | 1.0606 | 1.76 | 0.3174 |
| | | Ours | 29.56 | 0.6918 | **0.4916** | 0.8586 | 0.7368 | **2.55** | 0.3055 |
| | | Ours Random | 29.75 | 0.7302 | 0.4406 | **0.8835** | 0.7866 | 2.37 | 0.3083 |
| | **Flux.1-dev** | Orig | 40.22 | 0.8086 | 0.2212 | 0.7678 | 1.0844 | 1.68 | 0.3042 |
| | | Orig ($\omega_{CFG}=2$) | 35.89 | **0.8292** | 0.2934 | **0.8114** | **1.1465** | 1.73 | 0.3014 |
| | | CADS | 44.83 | 0.7321 | 0.3326 | 0.7192 | 0.9499 | **2.43** | 0.2656 |
| | | SPARKE | 37.75 | 0.8066 | 0.2578 | 0.7816 | 1.050 | 1.82 | **0.3067** |
| | | Ours | **34.93** | 0.8082 | 0.2941 | 0.7998 | 1.1356 | 1.87 | 0.3025 |
| | | Ours Random | 35.00 | 0.7854 | **0.3894** | 0.7992 | 1.0006 | 2.20 | 0.3005 |
| | **SANA1.5** | Orig | 51.53 | 0.7060 | 0.2040 | 0.5700 | 0.7350 | 1.60 | 0.3105 |
| | | Orig ($\omega_{CFG}=2$) | 53.40 | **0.7286** | 0.3150 | 0.6467 | **0.8328** | 1.81 | 0.3039 |
| | | CADS | 51.29 | 0.6992 | 0.3410 | 0.6104 | 0.5330 | 1.86 | 0.3064 |
| | | SPARKE | **42.72** | 0.6890 | 0.2490 | 0.6350 | 0.7365 | 1.67 | **0.3148** |
| | | Ours | 50.15 | 0.6943 | **0.4553** | 0.6726 | 0.5255 | **2.08** | 0.3076 |
| | | Ours Random | 47.20 | 0.6742 | 0.3784 | **0.6906** | 0.7207 | 1.94 | 0.3084 |

# 4. Experiments

We validate the effectiveness of DAVE on representative models, including Stable Diffusion 3.5 (Esser et al., 2024), FLUX.1-dev, and SANA1.5 (Xie et al., 2025), comparing our results against state-of-the-art diversity-enhancing samplers such as CADS (Sadat et al., 2023), Particle Guidance (PG) (Corso et al., 2023), SPARKE (Jalali et al., 2026),

SPELL (Kirchhof et al., 2024), DiverseFlow (Morshed & Boddeti, 2025), and Oscar (Wu et al., 2025). Using prompts sampled from ImageNet (Deng et al., 2009) and MS-COCO (Lin et al., 2014), we quantify performance across three primary dimensions: visual quality (FID, Precision, Density), generation diversity (Recall, Coverage, Vendi Score), and text-image alignment (CLIP score) (Heusel et al., 2017; Kynkäänniemi et al., 2019; Naeem et al., 2020;

A blue shelving unit has a vase and metal cups on it

A close up of a piece of cake on a plate

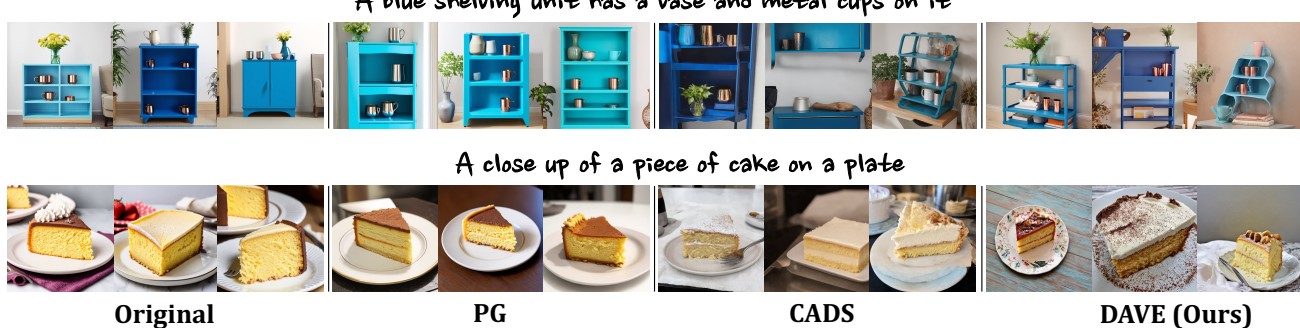

| **Original** | **PG** | **CADS** | **DAVE (Ours)** |

*Figure 4.* Qualitative comparison of in-batch diversity methods on Stable Diffusion 3. DAVE produces diverse samples with varied layouts and object appearances while preserving prompt consistency and visual fidelity.

Friedman & Dieng, 2022). Detailed implementation settings are presented in Appendix A.

### 4.1. Main Results

The results in Table 1 highlight the effectiveness of our approach. We quantitatively compare DAVE against (i) the original baseline, (ii) a low-CFG baseline ($\omega_{CFG}$=2, a standard heuristic for trading fidelity for diversity), and (iii) prior diversity-enhancement methods CADS and SPARKE (selected for their independence from in-batch settings). For evaluation, we generate 10 samples per prompt for 1,000 ImageNet class labels and 500 MS-COCO prompts. To demonstrate DAVE's robustness to block selection, we report results under both fixed-block and random-block settings. The fixed-block setting applies the intervention to a single predetermined layer, whereas the random-block setting dynamically samples from the identified pool $\mathcal{L}$. Although the random setting occasionally yields higher diversity, we adopt the fixed-block approach as our default in subsequent experiments to ensure strict reproducibility.

Our results demonstrate that DAVE consistently improves key diversity metrics across all three foundation models, outperforming the original sampler while remaining highly competitive with strong baselines. Crucially, these gains hold under both fixed and randomized block settings, confirming that our method is robust to the exact choice of intervention layer. Notably, while CADS achieves high diversity scores on ImageNet with Flux.1-dev and SANA1.5, it exhibits a severe degradation in precision and CLIP alignment. This suggests that its conditioning-perturbation strategy may be brittle when applied to certain flow-based architectures. In contrast, DAVE delivers substantial diversity improvements while reliably preserving text alignment and image quality, ultimately yielding a favorable diversity–fidelity trade-off (See Appendix C, Figure 11).

*Table 2.* Quantitative results for in-batch diverse generation on Stable Diffusion 3.5.

| Method | FID | Prec | Rec | Vendi | CLIP |
|---|---|---|---|---|---|
| Orig | 36.37 | 0.821 | 0.255 | 2.294 | 0.313 |
| PG | 45.69 | 0.833 | 0.204 | 1.946 | 0.314 |
| SPELL | 37.13 | 0.816 | 0.235 | 1.699 | 0.318 |
| DiverseFlow | 39.78 | 0.818 | 0.209 | 1.557 | 0.312 |
| Oscar | 35.87 | 0.827 | 0.245 | 1.698 | 0.312 |
| **Ours** | **29.75** | 0.780 | 0.441 | 2.370 | 0.308 |

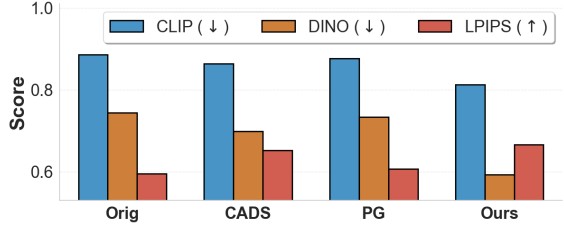

*Figure 5.* In-batch similarity comparison (batch size = 4). "Orig" denotes results from the vanilla Stable Diffusion 3.

We further evaluate the effectiveness of our method in the explicit in-batch generation setting. For this evaluation (Table 2), we compare DAVE against various recent baselines specifically designed for within-batch diversity: SPELL, DiverseFlow, OSCAR, and Particle Guidance (PG). Evaluated on Stable Diffusion 3 and 3.5 (batch size = 4) using MS-COCO prompts, DAVE consistently outperforms these baselines across all diversity metrics while incurring only marginal drops in image quality. Furthermore, as analyzed in Figure 5, DAVE achieves substantially lower feature-level similarity (CLIP/DINO) and higher perceptual distance (LPIPS), confirming its robust capability to maximize intra-batch diversity. Finally, qualitative results (Figure 4) corroborate these findings: while CADS offers competitive

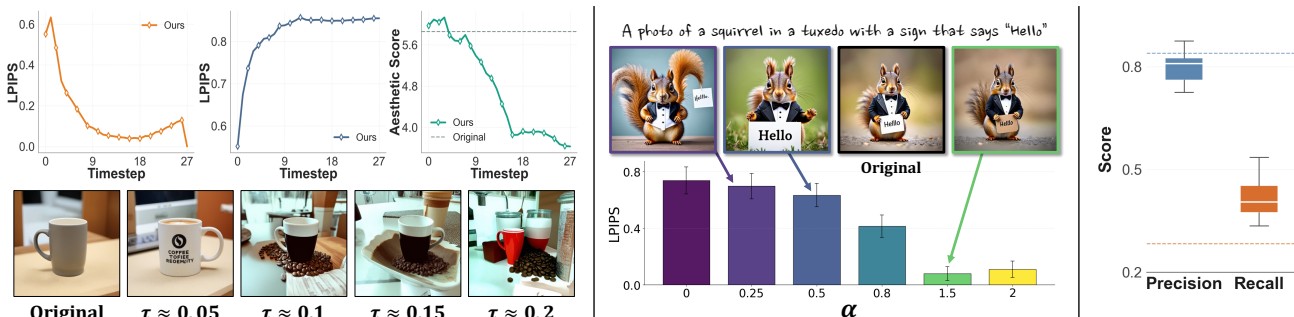

*Figure 6.* Ablations on hyperparameters. (Left) Effect of the temporal cutoff $\tau$ on the magnitude of image change and quality. (Middle) Image-change magnitude as a function of the attenuation strength $\alpha$. (Right) Robustness to block selection within the block pool $\mathcal{L}$.

visual variation but frequently compromises prompt adherence, DAVE consistently generates diverse, high-fidelity images that strictly respect the text condition.

## 4.2. Ablation Studies

In this section, we conduct a series of ablation studies to validate our parameter selections and provide deeper insights into the dynamics of DC attenuation.

**Temporal Cutoff $\tau$.** We evaluate the sensitivity of DAVE to the temporal cutoff $\tau$, which restricts the intervention to the initial $\lfloor \tau \cdot T \rfloor$ steps (where $T$ denotes the total number of timesteps). Consistent with our hypothesis, the results indicate that trajectory lock-in is indeed concentrated within the earliest stages of the generative process (Figure 6-Left). To further investigate this, we conducted a timestep-wise analysis by partitioning the denoising trajectory into early, middle, and late phases (5 steps each), as summarized in Table 3. Our findings show that cross-seed DC similarity exhibits a clear decreasing trend toward later timesteps. Additionally, the DC energy ratio is highest in the early regime, where our intervention triggers the most significant structural divergence. While accumulating interventions over more timesteps increases structural variation, it eventually introduces a trade-off with image quality. Notably, at the recommended threshold of $\tau = 0.15$ (approximately the first 4 steps in Stable Diffusion 3), the model maintains its structural integrity and even shows a slight improvement in aesthetic scores compared to the unperturbed baseline. This suggests that early-stage DC attenuation successfully promotes diversity without compromising the subsequent refinement of local details.

**Attenuation Strength $\alpha$.** The parameter $\alpha$ regulates the intensity of structural manipulation. As shown in Figure 6, lower $\alpha$ values yield higher LPIPS scores, indicating greater structural deviation from the baseline generation. While more aggressive attenuation enhances diversity, settings

*Table 3.* Step-wise analysis of DC characteristics and perceptual impact on SD3. **CS-Sim**, **ER**, and **LPIPS** denote Cross-seed DC similarity, DC energy ratio, and LPIPS distance, respectively.

| Steps | CS-Sim | ER | LPIPS |
|---|---|---|---|
| Early (0–4) | 0.975 | 0.512 | 0.717 |
| Mid (12–16) | 0.893 | 0.304 | 0.113 |
| Late (24–27) | 0.756 | 0.298 | 0.095 |

below $\alpha \approx 0.2$ can compromise structural integrity. Interestingly, we observe an asymmetric response: amplifying the DC component ($\alpha > 1$) results in negligible structural changes. This validates our interpretation of the DC mode as a structural anchor—its suppression releases the model from trajectory lock-in, whereas further amplification imposes no additional constraints on the already established generative pathways. Further sensitivity analyses are detailed in Appendix B.

**Target Block Pool $\mathcal{L}$.** We identify the target pool $\mathcal{L}$ by isolating Transformer blocks that exhibit a high degree of cross-seed DC invariance. This phenomenon indicates a deterministic bottleneck, where internal DC components remain nearly identical despite differing initial noise realizations. To objectively quantify this alignment, we apply a $t$-test with Benjamini-Hochberg FDR correction (Benjamini & Hochberg, 1995) ($p < 0.05$) to blocks maintaining a cross-seed cosine similarity of $\geq 0.99$. In Stable Diffusion 3, this criterion yields a specific subset: $\mathcal{L} = \{0, 2, 4, 5, \dots, 17\}$. Empirically, applying DAVE to any candidate block within $\mathcal{L}$ yields a consistent boost in Recall with only a marginal reduction in Precision (Figure 6-Right). These findings confirm that early-stage DC alignment is a critical prerequisite for the intervention to successfully break trajectory lock-in. By explicitly targeting this representational property, DAVE establishes a stable framework for diversity enhancement that is highly robust to block selection.

*Table 4.* Cross-dataset robustness of the $\mathcal{L}$ selection strategy for diversity enhancement.

| Method | Vendi (↑) | CLIP (↑) |
|---|---|---|
| Original | 1.4541 | 0.3088 |
| DAVE | **2.0275** | 0.3061 |
| Cross-dataset | 1.8840 | 0.3061 |

We further examined the robustness of the target block pool $\mathcal{L}$ with respect to the calibration prompt set. To test this, we compared pools independently constructed from 20 randomly sampled MS-COCO captions and 20 ImageNet labels. In Stable Diffusion 3.5, these independently derived pools exhibited near-complete overlap, achieving an average pairwise Jaccard similarity of 0.98. We also conducted a cross-dataset evaluation by applying the pool $\mathcal{L}$ identified from one source to the other; the diversity gains remained robust, showing only a marginal performance gap compared to the in-dataset setting (Table 4). Together, these results demonstrate the intrinsic robustness of our $\mathcal{L}$ selection strategy. Consequently, this unified rule—defining the target pool via early-stage cross-seed DC alignment—is inherently dataset-agnostic, enabling seamless adaptation across diverse domains with a single, one-time calibration rather than exhaustive per-dataset tuning.

## 5. Discussion

### 5.1. Trajectory Analysis

We empirically verify that DAVE mitigates trajectory lock-in, allowing generative trajectories to evolve along more diverse paths. To evaluate sample-to-sample separation, we compare the pairwise cosine distances along the generative trajectory between baseline SD3 samples and those generated with DAVE. As shown in Figure 7 (Right), DAVE leads to a progressive increase in the average inter-sample distance as the generation proceeds. This indicates that our early-stage intervention effectively expands the accessible state space, facilitating richer structural branching in later steps.

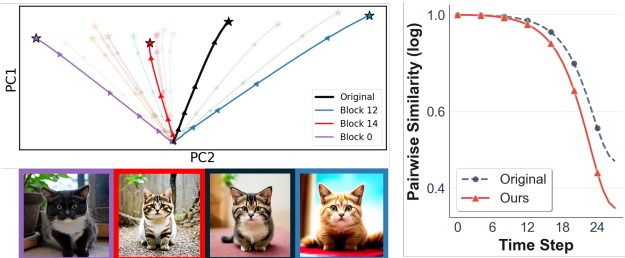

*Figure 7.* (Left) Latent trajectory visualization with generated outputs. (Right) Pairwise latent similarities across steps.

Furthermore, DAVE provides structural flexibility through selective intervention across Transformer blocks $\ell \in \mathcal{L}$, enabling entirely diverse outcomes from identical initial noise. As qualitatively confirmed by the visualized trajectories and resulting images in Figure 7 (Left), DAVE effectively prevents premature convergence to a single deterministic path, fostering a much broader and more expansive trajectory evolution.

*Table 5.* Quantitative results with DAVE in SD3.5-Large-Turbo.

| Method | Precision | Recall | Vendi | CLIP |
|---|---|---|---|---|
| Orig | **0.816** | 0.384 | 1.383 | 0.205 |
| CADS | 0.785 | 0.423 | 1.565 | 0.206 |
| PG | 0.788 | 0.418 | 1.422 | 0.211 |
| Ours | 0.798 | **0.431** | **1.643** | 0.205 |

### 5.2. DAVE on Distilled Models

A notable advantage of DAVE is its inherent compatibility with distilled models. Because the method intervenes directly at the level of internal representations rather than modifying the scheduler or sampling rules, it is straightforward to apply even when the sampling procedure is heavily streamlined by distillation. We validate this capability on the distilled model SD3.5-Large-Turbo (Esser et al., 2024), where DAVE consistently improves image diversity, as demonstrated by both quantitative metrics (Table 5) and qualitative visualizations (Figure 8).

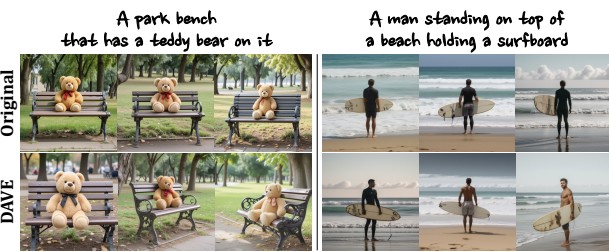

*Figure 8.* Qualitative results with DAVE in SD3.5-Large-Turbo.

### 5.3. Block-wise Analysis

In further analysis of DAVE, we examine block-wise manipulation and find a consistent tendency: some blocks repeatedly induce diversity along specific attribute directions. We quantify these shifts by scoring changes relative to the original images across three attributes—Color, Size, Texture—using Gemini 3 Flash. On Stable Diffusion 3.5, structural divergence (DAVE's primary objective) is reliably achieved across blocks, yet attribute changes are block-dependent: Blocks 1–3 predominantly vary color, Block 0 often alters subject scale/size, and Block 14 tends to pro-

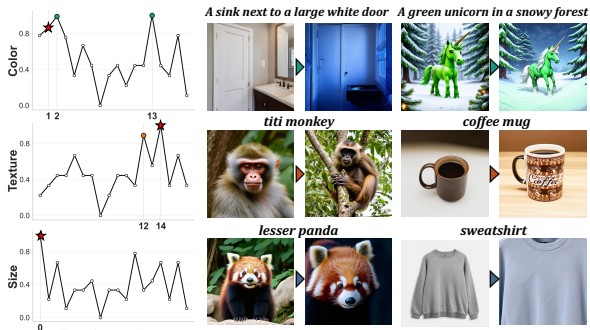

*Figure 9.* Block-wise change direction analysis for DAVE in Stable Diffusion 3. (Left) VLM-based attribute scores quantifying the change direction for each block. (Right) Representative output examples showing the block-specific changes.

duce coarser texture. Additional model cases are deferred to Appendix F.1.

These findings suggest that DC attenuation may interact with internal blocks in a block-dependent manner, offering a mechanistic perspective on how diversity emerges from localized representation-level interventions. Although inducing attribute-specific diversity is not a primary objective of our method, the observed block signatures indicate a potential route to controllable diversity when paired with deeper architectural insight.

*Table 6.* Comparison between Original baseline and DAVE across different CFG scales on SD3. (default $\omega_{CFG} = 7$)

| CFG | Method | Prec | Rec | CLIP | Vendi |
|---|---|---|---|---|---|
| 3 | Orig | 0.860 | 0.052 | 0.313 | 2.540 |
| | Ours | 0.752 | 0.165 | 0.310 | 2.307 |
| 5 | Orig | 0.883 | 0.016 | 0.314 | 2.975 |
| | Ours | 0.795 | 0.133 | 0.312 | 2.995 |
| **7** | Orig | 0.871 | 0.015 | 0.316 | 3.385 |
| | Ours | 0.735 | 0.132 | 0.313 | 3.828 |
| 10 | Orig | 0.854 | 0.035 | 0.318 | 4.189 |
| | Ours | 0.674 | 0.162 | 0.307 | 5.370 |

### 5.4. Compatibility with Classifier-Free Guidance

DAVE is compatible with CFG and can be naturally combined with it to further enhance performance. Across all CFG scales, DAVE substantially improves Recall, and at moderate-to-high CFG scales it also improves Vendi, while keeping CLIP nearly unchanged. This is practically important because higher CFG typically improves fidelity at the cost of diversity, whereas DAVE helps recover diversity under such settings without harming text alignment. Note

that our main experiments already follow the default CFG setting of each base model (Table 8), showing that DAVE works under standard practical configurations.

### 5.5. DAVE with Highly Constrained Prompts

We evaluated DAVE on PartiPrompts (Yu et al., 2022), a benchmark categorized by varying complexity levels. Although complex prompts naturally allow less variation than simpler ones, DAVE consistently improves Vendi-score over the original model across all prompt complexity levels. Notably, CLIP-score changes remain minimal (absolute gap $\leq 0.01$), demonstrating that DAVE enhances meaningful diversity without compromising prompt alignment, even under high structural constraints.

*Table 7.* Evaluation across prompt complexities. Bold indicates the best performance in each metric.

| Metric | Method | Basic | Simple | Fine | Comp. |
|---|---|---|---|---|---|
| Vendi | Original | 2.051 | 1.575 | 1.524 | 1.511 |
| | DAVE | **2.466** | **2.023** | **2.009** | **2.006** |
| CLIP | Original | 0.288 | **0.344** | **0.331** | **0.321** |
| | DAVE | **0.298** | 0.337 | 0.329 | 0.311 |

### 5.6. Computational Efficiency of DAVE

DAVE uses minimal computational overhead while significantly improving generation diversity. Unlike prior approaches that rely on additional optimization, feature-bank maintenance, or explicit inter-sample interactions during sampling, DAVE performs lightweight representation-level modulation directly on intermediate representations during the early generation stage. Detailed complexity analyses are provided in Appendix G.

## 6. Conclusion

In this paper, we addressed the critical issue of limited generation diversity in flow-based text-to-image models, identifying the rapid homogenization of the DC component as a primary bottleneck. To mitigate this, we introduced DC Attenuation for diVersity Enhancement (DAVE), a streamlined representation-level intervention. Our extensive experiments demonstrate that DAVE effectively broadens the generative range, yielding diverse images without compromising prompt alignment, thereby maintaining a highly favorable diversity–fidelity trade-off. As a training-free method with negligible overhead, DAVE offers a practical and scalable framework for unlocking generative models from premature structural lock-in.

## Acknowledgements

This work was supported by the Institute for Information & Communications Technology Planning & Evaluation (IITP) grant funded by the Korea government (MSIT) (RS-2019-II190075, Artificial Intelligence Graduate School Support Program (KAIST); RS-2024-00457882, AI Research Hub Project; RS-2022-II220984, Development of Artificial Intelligence Technology for Personalized Plug-and-Play Explanation and Verification of Explanation) and by the InnoCORE program of the Ministry of Science and ICT (N10250156).

## Impact Statement

This paper presents work whose goal is to advance the field of Machine Learning. There are many potential societal consequences of our work, none of which we feel must be specifically highlighted here.

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

# A. Implementation Details

**Models and Datasets.**    We conduct experiments on four representative flow-based text-to-image generation models: Stable Diffusion 3 (SD3), Stable Diffusion 3.5 (SD3.5), FLUX.1-dev, and SANA1.5. All methods are evaluated on ImageNet and MS-COCO benchmarks. For ImageNet, we generate 10 samples for each of the 1,000 class-label prompts, resulting in 10K images per method. For MS-COCO, we randomly sample 500 captions from the val2017 split and generate 10 images per prompt, yielding a total of 5K images.

**Evaluation Metrics.**    We evaluate performance along three dimensions: visual quality, diversity, and text–image alignment. Visual quality and diversity is measured using FID, Precision, Recall, Coverage, and Density, computed in a shared Inception feature space. For ImageNet, $N_{\text{fake}} = 10\text{K}$ generated images are compared against $N_{\text{real}} = 10\text{K}$ real images from the corresponding validation split. For MS-COCO, we similarly use $N_{\text{fake}} = 5\text{K}$ generated images and $N_{\text{real}} = 5\text{K}$ real images.

Diversity is assessed using the Vendi Score (Friedman & Dieng, 2022), computed on the cosine-similarity Gram matrix of L2-normalized CLIP ViT-B/32 image embeddings (Radford et al., 2021). Text–image alignment is evaluated using the CLIP Score (Hessel et al., 2021), computed as the cosine similarity between CLIP ViT-B/32 image and text embeddings. Both metrics are computed independently for each prompt and reported as averages across all prompts.

## A.1. Experimental Settings

### A.1.1. BLOCK SELECTION

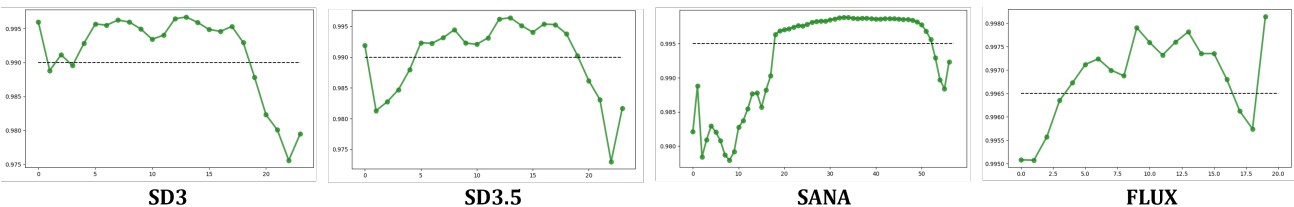

| SD3 | SD3.5 | SANA | FLUX |
|---|---|---|---|

*Figure 10.* Mean cosine similarity of DC components of hidden representations $h_t^{(\ell)}$ across 100 random noise seeds, measured at each Transformer block. Higher similarity indicates stronger cross-seed convergence.

To identify suitable intervention blocks for DAVE, we measure the mean cosine similarity of DC components across 100 different noise seeds at each Transformer block. As shown in Figure 10, early and intermediate blocks exhibit consistently high cosine similarity, indicating strong cross-seed alignment and pronounced representational lock-in.

**Block Selection Rationale.**    Based on this analysis, we select intervention blocks from regions where the DC component becomes nearly seed-invariant. For SD3, blocks in the set $\mathcal{L} \in \{0, 2, 4, 5, \ldots, 17\}$ exhibit high cosine similarity and are used as candidate blocks in the random-block setting. For SD3.5, a similar trend is observed for $\mathcal{L} \in \{0, 5, \ldots, 15\}$. In the fixed-block configuration for both models, we select a representative block ($\mathcal{L} = 5$) within this high-alignment regime.

For FLUX.1-dev and SANA1.5, high DC cosine similarity persists over a broader range of blocks, with peak alignment occurring at later blocks compared to SD-based models. Accordingly, we select $\mathcal{L} = 30$ for FLUX.1-dev and $\mathcal{L} = 13$ for SANA1.5 in the fixed-block setting, and define model-specific candidate pools ($\mathcal{L} \in [19, 40]$ for FLUX.1-dev and $\mathcal{L} \in [4, 16]$ for SANA1.5) for the random-block configuration.

**Statistical Criterion.**    To quantify cross-seed invariance of DC components, we compute pairwise cosine similarity across 100 noise seeds at each block. Candidate blocks are identified using a two-sided $t$-test with Benjamini–Hochberg FDR correction (Benjamini & Hochberg, 1995) ($p < 0.05$), and we further require the mean cosine similarity to exceed 0.99. Although this criterion identifies a broad set of high-alignment blocks, we empirically observe that interventions applied to final-stage blocks have limited impact on generation diversity. These late blocks are therefore excluded from the candidate pools.

*Table 8.* DAVE parameter settings for quantitative evaluation. "$\omega_{CFG}$" denotes the default classifier-free guidance scale of each base model, used in all main experiments unless stated otherwise.

| Model | Setting | $\omega_{CFG}$ | $\tau$ | $\mathcal{L}$ | $\alpha$ |
|---|---|---|---|---|---|
| SD3 | Fixed | 7.0 | 0.15 | 5 | 0.5 |
| | Random | 7.0 | 0.15 | 0,2,4–17 | 0.5 |
| SD3.5 | Fixed | 7.0 | 0.15 | 5 | 0.5 |
| | Random | 7.0 | 0.15 | 0,5–15 | 0.5 |
| FLUX.1-dev | Fixed | 3.5 | 0.2 | 30 | 0.2 |
| | Random | 3.5 | 0.2 | 19–40 | 0.2 |
| SANA1.5 | Fixed | 4.5 | 0.2 | 13 | 0.2 |
| | Random | 4.5 | 0.2 | 4–16 | 0.2 |

### A.1.2. EVALUATION SETTINGS

**Quantitative Evaluation Settings**  For quantitative evaluation, we consider two configurations: a fixed-block setting and a random-block setting. In the fixed-block setting, DC attenuation is applied to a single representative block selected from the high-alignment regime. In the random-block setting, intervention blocks are uniformly sampled from model-specific candidate pools identified by the block selection analysis. This configuration is designed to demonstrate that DAVE is not tied to a specific block, but remains effective across diverse blocks satisfying the proposed criteria. The complete parameter settings are summarized in Table 8.

**In-batch Diversity Evaluation Settings**  For in-batch diversity evaluation, we use 200 randomly sampled captions from the MS-COCO val2017 split. For each prompt, we generate images using 5 fixed random seeds, and for each seed we produce 4 samples, resulting in a total of 4K generated images. To ensure fair comparison, the same set of random seeds is used for all methods. We compare the original sampler, CADS, Particle Guidance (PG), SPELL, DiverseFlow, OSCAR, and our method. In-batch diversity is measured using CLIP similarity, DINO feature similarity, and LPIPS, where lower similarity values indicate higher diversity within a batch.

## B. Sensitivity Analysis

*Table 9.* Sensitivity analysis of DAVE across hyperparameters $\alpha$ (intervention strength) and $\tau$ (intervention window). Metrics include Precision (Prec), Recall (Rec), Vendi score (Vendi), and CLIP score (CLIP).

| Effect of Intervention Strength ($\alpha$) | | | | | Effect of Intervention Window ($\tau$) | | | | |
|---|---|---|---|---|---|---|---|---|---|
| $\alpha$ | **Prec** | **Rec** | **Vendi** | **CLIP** | $\tau$ | **Prec** | **Rec** | **Vendi** | **CLIP** |
| 0.3 | 0.664 | 0.561 | 2.42 | 0.302 | 0.00 | 0.821 | 0.254 | 2.29 | 0.313 |
| 0.4 | 0.676 | 0.554 | 2.62 | 0.302 | 0.10 | 0.698 | 0.456 | 2.53 | 0.307 |
| **0.5** | **0.692** | **0.492** | **2.55** | **0.306** | **0.15** | **0.692** | **0.492** | **2.55** | **0.306** |
| 0.6 | 0.723 | 0.461 | 2.41 | 0.308 | 0.20 | 0.622 | 0.562 | 2.61 | 0.303 |
| 0.7 | 0.783 | 0.407 | 2.36 | 0.312 | 0.25 | 0.588 | 0.583 | 2.63 | 0.299 |

DAVE operates in a stable regime with an explicit diversity–fidelity trade-off, rather than as a brittle single-point intervention. The added ablations show that nearby tested settings around our practical defaults ($\alpha$=0.5, $\tau$=0.15) do not trigger abrupt quality collapse across any evaluated metrics, but instead produce a gradual trade-off between Precision/CLIP and Recall/Vendi. We therefore view $\alpha$ and $\tau$ as coarse intervention knobs controlling the strength and duration of early DC attenuation, rather than fragile hyperparameters that require fine-grained tuning.

## C. Trade-off Analysis

Figure 11 presents a Pareto analysis between diversity and fidelity metrics across different diversity-enhancement methods on SD3 and SD3.5. We compare the trade-off trajectories of the original sampler, CADS, SPARKE, and DAVE by jointly visualizing diversity (Vendi score) against perceptual quality (Aesthetic score) and semantic alignment (CLIP score). Each trajectory is obtained by sweeping the diversity-control parameter of each method, where points correspond to different attenuation steps in DAVE, corruption strengths in CADS, and guidance frequencies in SPARKE.

As shown in the figure, existing methods often improve diversity at the cost of image fidelity or text–image alignment, leading to unfavorable trade-offs. In contrast, DAVE achieves substantially higher diversity while preserving competitive aesthetic quality and semantic consistency. Moreover, DAVE exhibits smooth and stable trajectory transitions across attenuation strengths, indicating controllable diversity enhancement without severe structural degradation.

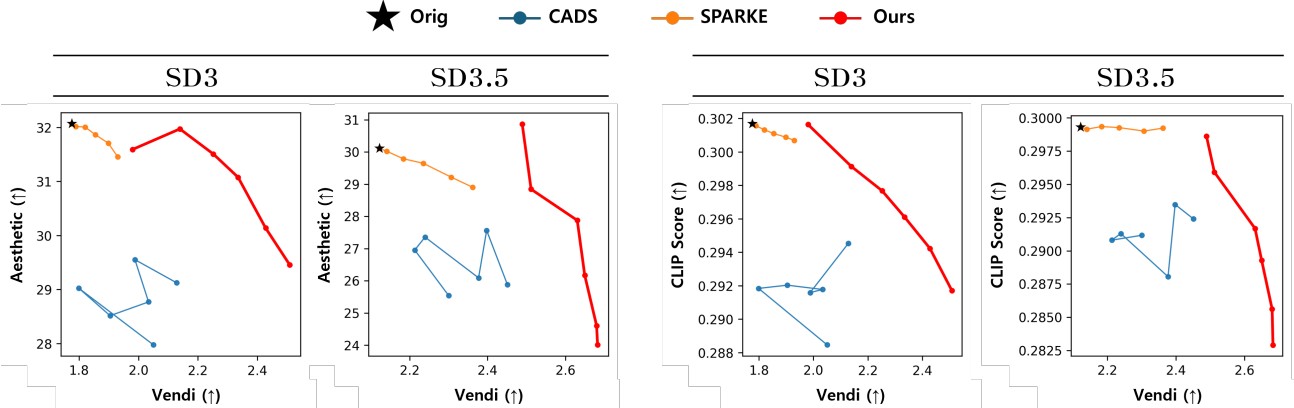

*Figure 11.* Pareto analysis between diversity and fidelity metrics. Compared to prior methods, DAVE achieves a more favorable trade-off frontier, improving diversity while preserving image quality and semantic alignment across SD3 and SD3.5.

## D. Pseudo Code

The pseudocode for DAVE is provided in Algorithm 1. The official implementation is available at `https://github.com/daheekwon/DAVE`.

---

**Algorithm 1** DAVE: DC component Attenuation for diVersity Enhancement

---

**Input:** Hidden states $H \in \mathbb{R}^{D \times C}$, Current timestep $t$, Block index $\ell$
**Input:** Hyperparameters: Cutoff $\tau$, Target block pool $\mathcal{L}$, Strength $\alpha$
 1: $H \leftarrow \text{TransformerBlock}_\ell(H)$
 2: **if** $(t < \tau)$**and** $(\ell \in \mathcal{L})$ **then**
 3:     $\mu \leftarrow \frac{1}{D}\sum_{d=1}^{D} H_{d,:}$                      ▷ Compute spatial mean (DC component)
 4:     **for** $d = 1$ **to** $D$ **do**
 5:         $H_{d,:} \leftarrow H_{d,:} + (\alpha - 1) \cdot \mu$                 ▷ Attenuate DC component
 6:     **end for**
 7: **end if**
**Output:** Enhanced hidden states $H$

---

## E. Theoretical Motivation for Early DC Lock-in

We provide a formal motivation for the intuition behind our method for a Transformer-based Flow Matching model. We divide our analysis into two parts: first, we characterize a mechanism by which the DC component of intermediate representations can become seed-invariant in the early high-noise sampling regime ($t \approx 0$); second, we formalize why early contraction can limit later trajectory separation under locally Lipschitz dynamics.

## E.1. Problem Setup and Definitions

**Definition E.1** (Spaces and Mapping). Let $\mathcal{X}$ be the image space and $\mathcal{H} = \mathbb{R}^{D \times C}$ be the latent space, where $D$ is the number of tokens and $C$ is the channel dimension. We define an abstract representation map induced by a fixed block of the pre-trained generator, $\mathcal{E} : \mathcal{X} \to \mathcal{H}$, mapping a sample image $X \in \mathcal{X}$ to its block-level hidden representation $h \in \mathcal{H}$.

Here $X$ is the state $x_t$ evolved by the ODE in Eq 1, and $\mathcal{E}$ to a fixed block $\ell \in \mathcal{L}$ (Eq 3), so that $h = \mathcal{E}(x_t)$ is precisely the block-level representation $h_t^{(\ell)}$ analyzed in Figure 2 and Table 3. We suppress $\ell$ and write $h_t := \mathcal{E}(x_t)$; the separation bounds below thus concern these representations across seeds.

**Definition E.2** (Distributions and Coupling). We consider the Flow Matching objective defined over two distributions:

- **Data Distribution** ($X_1$): $X_1 \sim p_{\text{data}}(X|c)$, conditioned on text $c$. The corresponding latent target is $h_1 = \mathcal{E}(X_1)$.

- **Noise Distribution** ($X_0$): $X_0 \sim p_{\text{noise}}(X)$. The corresponding latent noise is $h_0$.

We adopt the **1-Rectified Flow** framework (Liu et al., 2022), which constructs a straight-line probability path between distributions. In the standard training setup (prior to any reflow steps), the coupling between the source and target is **independent**, meaning the joint distribution factorizes as:

$$\pi(X_0, X_1) = p_{\text{noise}}(X_0) p_{\text{data}}(X_1|c) \tag{6}$$

This independence implies that the initial noise sample $X_0$ contains no information about the target data sample $X_1$ ($X_0 \perp X_1$).

**Definition E.3** (Lipschitz Continuity of Target Estimation). Let $\Psi(h) = \mathbb{E}_{X_1}[\mathcal{E}(X_1) \mid h, c]$ be the conditional expectation. We assume that $\Psi$ is $K$-Lipschitz continuous with respect to the latent metric. That is, there exists a constant $K < \infty$ such that for any $h, h' \in \mathcal{H}$:

$$\|\Psi(h) - \Psi(h')\| \leq K\|h - h'\| \tag{7}$$

This is a standard assumption in generative modeling to ensure the well-posedness of the induced ODE flow (Liu et al., 2022) and is often enforced in deep neural networks via regularization techniques (Miyato et al., 2018).

**Definition E.4** (Spectral Decomposition in Latent Space). For any latent state $h \in \mathcal{H}$, we decompose it into a **DC component** and a **AC component**:

$$h = h_{DC} + h_{AC} \tag{8}$$

where $h_{DC} \triangleq \frac{1}{D} \mathbf{1}_D \mathbf{1}_D^\top h$ is the spatial mean, and $h_{AC} \triangleq h - h_{DC}$ is the spatial residual. Equivalently, $h_{DC} = \mathbf{1}_D \mu_t$, where $\mu_t \in \mathbb{R}^{1 \times C}$ is the spatial-mean (DC) vector of Eq 4; thus $h_{DC}$ is the broadcast of the main-text DC vector across all $D$ tokens, and $h_{AC} = h - \mathbf{1}_D \mu_t$.

## E.2. Mechanism of Early Spectral Collapse

In this section, we analyze why the learned vector field collapses to the spatial mean at $t \approx 0$.

**Proposition E.5** (Dominance of Ensemble Mean in High-Noise Regime). *For a sufficiently small time $t \approx 0$ (high-noise regime), the optimal vector field $v^*$ is dominated by the drift towards the Conditional Ensemble Mean. The deviation caused by instance-specific details is strictly bounded by $\mathcal{O}(t)$.*

*Proof.* Let the objective be the standard MSE loss. It is a fundamental property of the $L_2$ risk that its global minimizer $v^*$ is the conditional expectation of the target vector field (Bishop, 2006; Lipman et al., 2022):

$$v^*(h_t, t; c) = \mathbb{E}_{X_0, X_1}[\mathcal{E}(X_1) - \mathcal{E}(X_0) \mid h_t; c]. \tag{9}$$

We decompose the optimal field into its two conditional expectations,

$$v^*(h_t, t; c) = \Psi(h_t) - \Phi(h_t), \quad \Psi(h_t) = \mathbb{E}[\mathcal{E}(X_1) \mid h_t, c], \quad \Phi(h_t) = \mathbb{E}[\mathcal{E}(X_0) \mid h_t, c]. \tag{10}$$

We analyze the target term $\Psi$ below; the source term $\Phi$ is treated after the collapse argument, as it accounts for the residual seed-specific (AC) content.

In the early phase, the interpolated state is $h_t = (1 - t)h_0 + th_1$. We compare the estimation at time $t$ with the estimation at time 0 using the Lipschitz assumption:

$$\|\Psi(h_t) - \Psi(h_0)\| \leq K\|h_t - h_0\| = K\|t(h_1 - h_0)\|. \tag{11}$$

At $t = 0$, due to independent coupling ($h_0 \perp h_1$), the input $h_0$ provides no information about $X_1$, so $\Psi(h_0) = \mathbb{E}[\mathcal{E}(X_1) \mid c] \triangleq \mu_c^H$. Thus, we can write:

$$\Psi(h_t) = \mu_c^H + R(t), \quad \text{where } \mathbb{E}[\|R(t)\|] \leq t \cdot K\, \mathbb{E}[\|h_1 - h_0\|]. \tag{12}$$

Thus, the expected magnitude of the residual is $\mathcal{O}(t)$. The residual $R(t)$ captures the seed- or instance-dependent correction required for recovering sample-specific structure. Since the conditional ensemble mean ($\mu_c^H$) remains an $\mathcal{O}(1)$ term as $t \to 0$, the early target estimate is dominated by this common component. $\square$

**Proposition E.6** (DC Dominance of the Conditional Ensemble Mean under Under-specified Prompts). *The Latent Ensemble Mean $\mu_c^H$ is spectrally sparse, dominated by the DC component ($h_{DC}$), as the spatial variations (AC components) cancel out across the data distribution.*

*Proof.* We apply the expectation operator to the spectral decomposition of the encoded target $\mathcal{E}(X_1)$:

$$\mu_c^H = \mathbb{E}_{X_1}[\mathcal{E}(X_1)_{DC} \mid c] + \mathbb{E}_{X_1}[\mathcal{E}(X_1)_{AC} \mid c]. \tag{13}$$

**1. Stability of DC:** The DC component $\mathcal{E}(X_1)_{DC}$ encodes global semantic information strongly correlated with the text $c$, aligning coherently across samples. Thus, $\|\mathbb{E}[\mathcal{E}(X_1)_{DC} \mid c]\| \gg 0$.

**2. Cancellation of AC:** The AC component $\mathcal{E}(X_1)_{AC}$ encodes high-frequency, spatially localized detail. We assume that, conditioned on $c$, the phase (spatial placement) of these details is not fully determined by the prompt but varies across the data distribution—e.g., a prompt fixes what objects appear and their coarse global statistics, but leaves their precise position, pose, and fine texture under-specified. Under this assumption, the AC components are not phase-aligned across samples, so they interfere destructively in expectation:

$$\mathbb{E}[\mathcal{E}(X_1)_{AC} \mid c] \approx 0. \tag{14}$$

Consequently, the learning target collapses to the spatial mean: $\mu_c^H \approx \mathbb{E}[\mathcal{E}(X_1)_{DC} \mid c]$. $\square$

**Summary of Collapse.** Together, Propositions E.5 and E.6 suggest that, in the early high-noise regime, the *target* term $\Psi$ is strongly biased toward a common DC-dominated direction $\mu_c^H$, while its seed-specific AC correction $R(t)$ remains $\mathcal{O}(t)$. The source term $\Phi$, by contrast, stays seed-dependent: at $t \approx 0$ we have $h_t \approx h_0$, so $\Phi$ reduces to the sample's own noise representation and retains its instance-specific (AC) content. Hence the early collapse is confined to the DC subspace of the target estimate, while AC variation is preserved—consistent with the empirically observed coexistence of high cross-seed DC similarity and low AC similarity in Figure 2. This provides a formal motivation for the observation that early trajectories become closely aligned in the DC subspace. The next result analyzes what happens once such early DC alignment has reduced the cross-seed DC separation to an $\epsilon$-neighborhood.

### E.3. Bounded Recovery after Early Lock-in

Having motivated why early DC alignment can reduce seed-specific separation in the DC subspace, we now show that any later recovery of this separation under locally Lipschitz ODE dynamics is bounded by the residual that remains after the early phase. Since the DC component encodes global layout and coarse structure—the under-specified factors that govern perceptual diversity—while the AC component carries localized texture, we treat the cross-seed separation in the DC subspace as the quantity controlling sample diversity.

Let $P_{DC} = \frac{1}{D}\mathbf{1}_D\mathbf{1}_D^\top$ denote the (linear, time-invariant) projection onto the DC subspace, so that $h_{DC} = P_{DC}h$. We strengthen Definition E.3 to the DC subspace: the learned field is assumed locally Lipschitz *on the DC subspace*, i.e. there exists $L < \infty$ such that for all states on the compact trajectory domain,

$$\|P_{DC}\big(v_\theta(x, t; c) - v_\theta(y, t; c)\big)\| \leq L\|P_{DC}(x - y)\|. \tag{15}$$

**Theorem E.7** (Bounded Diversity Recovery under ODE Dynamics). *Consider the ODE flow $\dot{h}_t = v_\theta(h_t, t; c)$ generated by the Transformer network, with $v_\theta$ locally Lipschitz on the DC subspace with constant $L < \infty$.[1] If the early spectral collapse (Section E.2) constrains the cross-seed DC separation to an $\epsilon$-neighborhood at an early time $t^*$, i.e. $\|P_{DC}(h_{t^*}^{(i)} - h_{t^*}^{(j)})\| \le \epsilon$, then the DC separation at the final time $t = 1$ is bounded by*

$$\|P_{DC}(h_1^{(i)} - h_1^{(j)})\| \le \epsilon \cdot \exp(L(1 - t^*)). \tag{16}$$

*Proof.* Let $h_t^{(i)}$ and $h_t^{(j)}$ be two trajectories starting from distinct noise samples whose DC separation is reduced to at most $\epsilon$ at an early time $t^*$ due to the mechanisms in Propositions E.5– E.6. Define the DC difference vector

$$\delta_{DC}(t) = P_{DC}(h_t^{(i)} - h_t^{(j)}). \tag{17}$$

For $t \ge t^*$, the time derivative of the squared distance is

$$\frac{d}{dt}\|\delta_{DC}(t)\|^2 = 2\langle \delta_{DC}(t), \dot{\delta}_{DC}(t)\rangle. \tag{18}$$

Since both trajectories evolve under the same vector field and text condition $c$, and $P_{DC}$ is linear and time-invariant,

$$\dot{\delta}_{DC}(t) = P_{DC}(v_\theta(h_t^{(i)}, t; c) - v_\theta(h_t^{(j)}, t; c)). \tag{19}$$

Using the Cauchy–Schwarz inequality and the DC-subspace Lipschitz condition,

$$\frac{d}{dt}\|\delta_{DC}(t)\|^2 \le 2\|\delta_{DC}(t)\| \, \|P_{DC}(v_\theta(h_t^{(i)}, t; c) - v_\theta(h_t^{(j)}, t; c))\| \le 2L\|\delta_{DC}(t)\|^2. \tag{20}$$

Applying Grönwall's inequality yields

$$\|\delta_{DC}(1)\|^2 \le \|\delta_{DC}(t^*)\|^2 \exp(2L(1 - t^*)). \tag{21}$$

Taking the square root and using $\|\delta_{DC}(t^*)\| \le \epsilon$, we obtain

$$\|P_{DC}(h_1^{(i)} - h_1^{(j)})\| = \|\delta_{DC}(1)\| \le \epsilon \exp(L(1 - t^*)). \tag{22}$$

This gives the desired bound. $\qquad\square$

Consequently, if the early collapse drives $\epsilon \to 0$, the right-hand side vanishes, so the final-time DC separation is upper-bounded by a quantity that tends to zero. This formalizes the lock-in intuition: once early dynamics align the global-structure (DC) subspace across seeds, later refinement can only amplify the residual DC differences that remain, rather than reconstructing the suppressed structural variation. We emphasize that this is a *capacity* bound—it shows that early DC alignment *caps* the recoverable diversity, which motivates intervening on the DC component early; the resulting diversity gains of DAVE are then established empirically.

# F. Additional examples

## F.1. Block-wise analysis in Flux

Here, we provide a complementary case study on Flux.1-dev to examine whether the block-dependent behaviors observed in Stable Diffusion extend to a different architecture. Beyond confirming that DC attenuation produces consistent seed-level divergence in the intended regime, we observe that the resulting attribute profile is again block-dependent, but the specific block–attribute associations are model-specific. Flux.1-dev exhibits systematic preferences toward certain attribute shifts, but these patterns do not map one-to-one to the block-index trends in Stable Diffusion 3.5, consistent with architectural and training differences. Figure 12 visualizes these Flux-specific tendencies—most prominently in texture and color—supporting the view that DC attenuation interacts with internal blocks in a structured yet model-dependent manner and may serve as a basis for controllable diversity when paired with architectural insight.

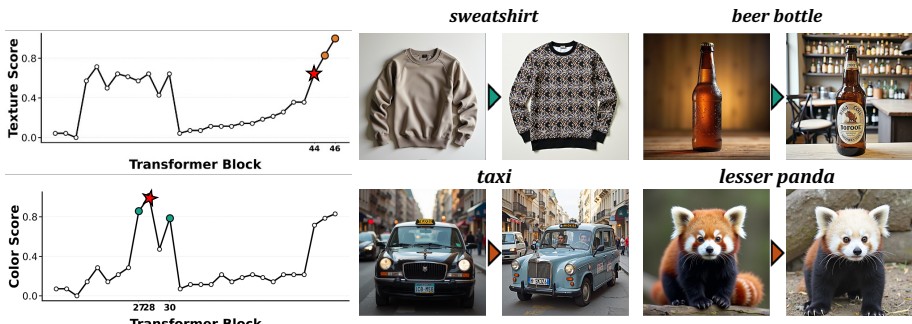

*Figure 12.* Block-wise Analysis on Flux.1-dev.

### F.2. Qualitative Examples

Following the results presented in the main text, we provide additional qualitative visualizations for each model in this section. To ensure a rigorous and consistent comparison, all images were generated using the same seeds as the baseline samples produced without our internal manipulations. This allows for a direct, side-by-side observation of how our method influences the generative process across various architectures while maintaining the fundamental structural characteristics of the original latent trajectories.

## G. Computational Cost

We analyze the computational costs of DAVE and compare it with existing diversity-enhancement methods. Let $T$ denote the number of sampling steps and $B$ the batch size.

**Baseline** diffusion or flow-based sampling requires one model evaluation per step, resulting in linear complexity with respect to both the number of sampling steps and the batch size: $O(TB)$.

**Lightweight perturbation-based methods**, such as CADS, preserve linear computational scaling by applying lightweight conditioning perturbations during sampling without explicit inter-sample interactions. Similarly, SPARKE maintains linear asymptotic scaling with respect to the batch size, although additional feature-statistics computations may introduce nontrivial practical memory overhead during sampling.

In contrast, **interaction-based diversity-enhancement methods**, including Particle Guidance (PG), DiverseFlow, OSCAR, and SPELL, require explicit or implicit interactions across samples during sampling. Particle Guidance and SPELL rely on inter-sample repulsion terms, DiverseFlow introduces kernel-based coupling across trajectories, and OSCAR performs batch-wise diversity optimization using feature-space interactions. Consequently, these methods may incur computational and memory overhead that scales quadratically with respect to the batch size due to batch-wise inter-sample interactions, resulting in overall complexity of $O(TB^2)$.

**DAVE** performs lightweight representation-level modulation only during the early denoising stage. By directly operating on intermediate representations without requiring feature-bank maintenance or batch-wise inter-sample interactions, DAVE preserves linear batch scaling and maintains near-identical practical runtime and memory usage compared to the baseline.

Table 10 summarizes the computational complexity of existing diversity-enhancement approaches. Lightweight perturbation-based methods preserve linear scaling with relatively small additional overhead, whereas interaction-based methods incur substantially larger computational and memory costs due to batch-wise diversity interactions during sampling. In contrast, DAVE maintains linear scaling with respect to the batch size through lightweight representation-level modulation without requiring feature-bank maintenance, inter-sample interactions, or additional optimization procedures.

As shown in Table 11, DAVE preserves near-identical runtime and reserved GPU memory usage to the original SD3 sampler

---

[1]**Justification for the Lipschitz Assumption:** While standard dot-product attention is not globally Lipschitz, practical implementations employ Layer Normalization and operate on bounded latent representations (compact support). Thus, the vector field is differentiable with a bounded gradient (finite $L$) along the integration path.

*Table 10.* Computational complexity of diversity-enhancement methods.

| Method | Asymptotic time | Additional memory | Batch scaling |
|---|---|---|---|
| Baseline | $O(TB)$ | $O(B)$ | Linear |
| CADS | $O(TB)$ | $O(B)$ | Linear |
| SPARKE | $O(TB)$ | $O(B)$ | Linear |
| Particle Guidance | $O(TB^2)$ | $O(B^2)$ | Quadratic |
| DiverseFlow | $O(TB^2)$ | $O(B^2)$ | Quadratic |
| OSCAR | $O(TB^2)$ | $O(B^2)$ | Quadratic |
| SPELL | $O(TB^2)$ | $O(B^2)$ | Quadratic |
| **DAVE (Ours)** | $\mathbf{O(TB)}$ | $\mathbf{O(B)}$ | Linear |

*Table 11.* Runtime and memory comparison of diversity-enhancement methods on SD3.

| Method | ms/img ↓ | img/s ↑ | Reserved Mem ↓ |
|---|---|---|---|
| | *Batch Size = 1* | | |
| Baseline | $2197.06 \pm 25.93$ | 0.455 | 21.0G |
| CADS | $2217.12 \pm 22.10$ | 0.451 | 21.0G |
| SPARKE | $2330.86 \pm 46.11$ | 0.429 | 37.7G |
| **DAVE (Ours)** | $2201.55 \pm 42.20$ | 0.454 | 21.0G |
| | *Batch Size = 4* | | |
| Baseline | $1956.97 \pm 4.75$ | 0.511 | 33.4G |
| SPELL | $2791.49 \pm 2.48$ | 0.358 | 34.3G |
| Particle Guidance | $9400.93 \pm 36.32$ | 0.106 | 49.4G |
| **DAVE (Ours)** | $1956.98 \pm 3.25$ | 0.511 | 33.4G |

while substantially outperforming interaction-based approaches in practical efficiency. In particular, DAVE avoids the significant computational and memory overhead associated with costly batch-wise inter-sample interactions during sampling. All measurements are averaged over 100 runs under identical sampling settings on a single NVIDIA H100 GPU.

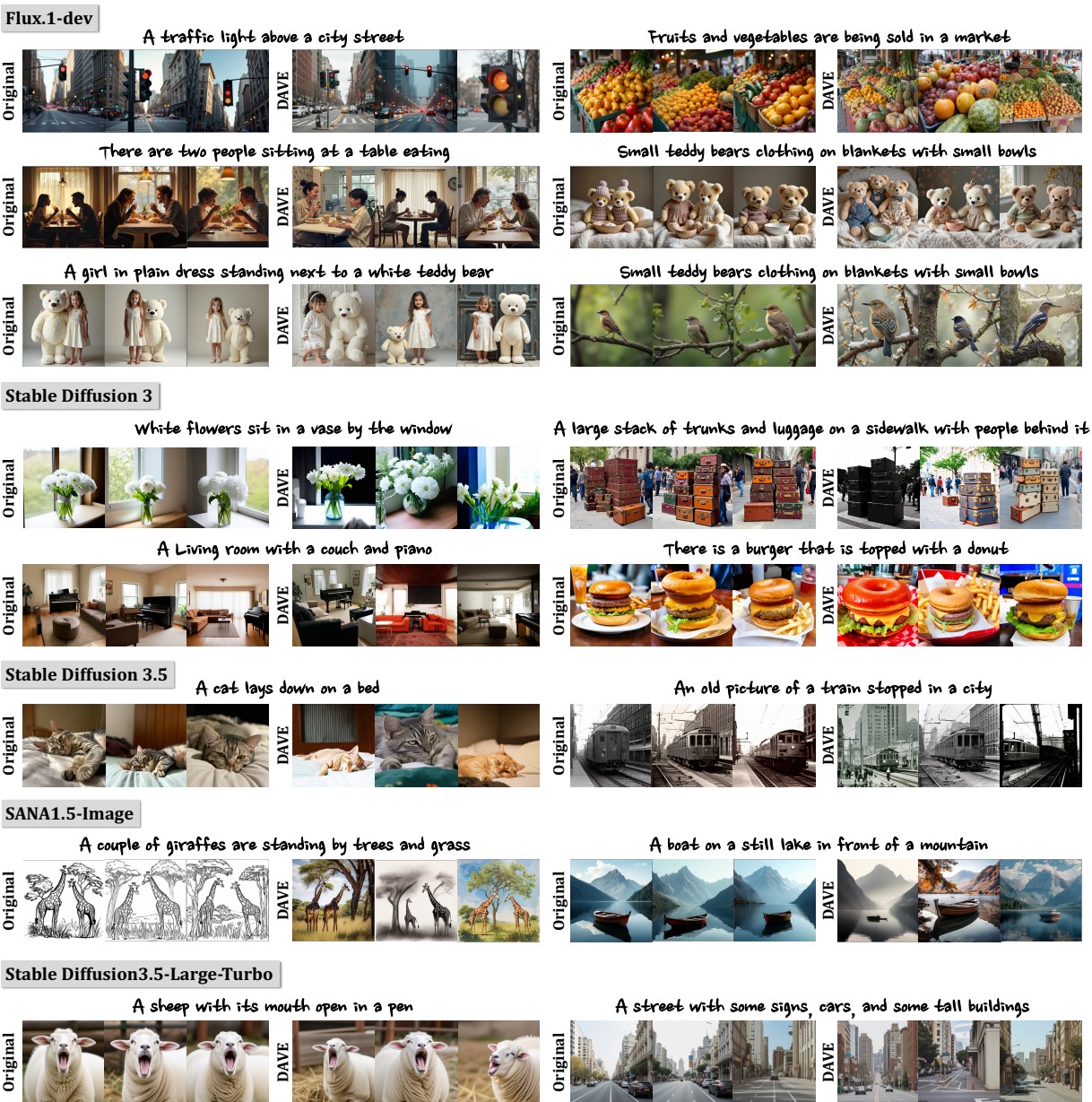

*Figure 13.* Qualitative results across different models.

