# OpenReview forum: "Breaking the Lock-in: Diversifying Text-to-Image Generation via Representation Modulation"
_ICML.cc/2026/Conference — ICML 2026 regular_

### Official Review · Reviewer_i8dp · 2026-03-08

**Soundness:** 3
**Presentation:** 2
**Significance:** 2
**Originality:** 3
**Overall Recommendation:** 4
**Confidence:** 4

**Summary:**

This paper is focused on increasing the generation diversity of a text-to-image diffusion model. The authors find that the low-frequency components of the denoiser model features in early denoising steps are correlated across different random seeds given the same text prompt. Motivated by this observation, the paper proposes to manually tuned down the low-frequency component to amplify the differences of these features across different seeds and consequently improve the generation diversity.

**Compliance With Llm Reviewing Policy:**

Affirmed.

**Final Justification:**

I have read the responses and the additional experiments have addressed my concerns on the hyperparameters and extra comparison. Therefore, I'd like to raise to rating to 4.

**Key Questions For Authors:**

I'm curious how does DC similarity across different seeds numerically change at different sampling timesteps. Do the authors have done any studies on that? I wonder if that is why the $\tau$ is limited a very small value.

**Limitations:**

Yes

**Strengths And Weaknesses:**

[Strengths]

1. The finding that low-frequency components of the features are shared across different random seeds is interesting and novel to the best of my knowledge.

2. The proposed method is manually reduce the magnitude of the low-frequency component is reasonable.

3. Ablations study done on the introduced hyperparameters are extensive and helps understand the effect of each individual choices.

[Weaknesses]

1. As manually modifying the intermediate features can easily introduce out-of-distribution features to the model, the proposed method is highly sensitive to the hyperparameters: a slightly different hyperparameters can greatly undermine the generation quality. As a result, different hypeparameters such as the block selection need to be applied when different base model is used.

2. The authors always use the spatial mean as the low-frequency component. It's analogous to an inference-time re-normalization of the features. The paper lacks analysis whether better preserving the feature statistics help improve the robustness of the method to the hyperparameters. For example, changing equation (5) to  $\hat{h}_t^{(\ell)} = \alpha \cdot \mu_t^{(\ell)} + (1-\alpha) \cdot \left( h_t^{(\ell)} - \mu_t^{(\ell)} \right)$  would help preserve the scale of the features.

3. The writing can still be improved with more proof-reading. For example, "infivsyinh" in line 025, "perturbationa" in line 033, "stablecue" in line 190, "broden" in line 173.

---

> ### Author Rebuttal · Authors · 2026-03-30
>
> We thank the reviewer for the thoughtful comments, especially regarding robustness, feature statistics, and timestep-wise behavior. We have carefully considered all feedback and incorporated detailed responses to each point within this rebuttal.
>
> ### **Sensitivity on hyperparameters ($\tau$ and $\alpha$)**
> - DAVE operates in a stable regime with an explicit diversity–fidelity trade-off, rather than as a brittle single-point intervention. The added ablations show that nearby tested settings around our practical defaults (𝛼=0.5, 𝜏=0.15) do not trigger abrupt quality collapse across any evaluated metrics, but instead produce a gradual trade-off between Precision/CLIP and Recall/Vendi. We therefore view 𝛼 and 𝜏 as coarse intervention knobs controlling the strength and duration of early DC attenuation, rather than fragile hyperparameters that require fine-grained tuning.
>
> | **$\alpha$** | Precision ($\uparrow$) | Recall ($\uparrow$) | Vendi ($\uparrow$) | CLIP ($\uparrow$) |
> | :--- | :---: | :---: | :---: | :---: |
> | 0.3 | 0.664 | 0.561 |2.42 | 0.302 |
> | 0.4 | 0.676 | 0.554 | 2.62 | 0.302 |
> | **0.5** | 0.692 | 0.492 | 2.55 | 0.306 |
> | 0.6 | 0.723 | 0.461 | 2.41 | 0.308 |
> | 0.7 | 0.783 | 0.407 | 2.36 | 0.312 |
>
> | **$\tau$** | Precision ($\uparrow$) | Recall ($\uparrow$) | Vendi ($\uparrow$) | CLIP ($\uparrow$) |
> | :--- | :---: | :--- | :---: | :---: |
> | 0.0 | 0.821 | 0.254 | 2.29 | 0.313 |
> | 0.1 | 0.698 | 0.456 | 2.53 | 0.307 |
> | **0.15** | 0.692 | 0.492 | 2.55 | 0.306 |
> | 0.2 | 0.622 | 0.562 | 2.61  |0.303 |
> | 0.25 | 0.588 |0.583 | 2.63 | 0.299 |
>
> - **Regarding block selection**, **$\mathcal{L}$** is determined by a fixed measurable criterion—early cross-seed DC alignment—rather than by arbitrary per-model heuristics. We further tested whether this calibration is prompt-set dependent, and both the prompt-source robustness analysis and cross-dataset transfer results support that the selected pool is not tied to a particular calibration set. Please refer to our response to *Reviewer uGg4* for the detailed results.
>
> ### **Comparison on Scale-preserving Variant**
> - We thank the reviewer for the suggestion of a scale-preserving variant. To examine this directly, we evaluated the modified form $\hat h = \alpha \mu + (1 - \alpha)(h - \mu)$ on the SD3.5 model across $\alpha \in \lbrace{0.2, 0.4, 0.6, 0.8\rbrace}$ using MSCOCO prompts. This variant yielded relatively stable LPIPS scores across $\alpha$, indicating that the intervention magnitude becomes less sensitive to the strength parameter. However, it exhibited lower Aesthetic and Vendi scores than our original formulation, particularly when $\alpha > 0.5$, a regime where the non-DC components are more heavily attenuated relative to the DC component. These results suggest that while scale-preserving formulations can enhance stability across different $\alpha$ values, they tend to sacrifice aesthetic quality and distributional diversity. Consequently, we maintain our current formulation as the primary method, while acknowledging scale-preserving variants as a promising direction for further enhancing the model's robustness.
>
>  | **α** | LPIPS (Ours)  | LPIPS (Scaled)  |Aesthetic (Ours)  |Aesthetic (Scaled) | CLIP (Ours)  | CLIP (Scaled)  | Vendi (Ours)  | Vendi (Scaled) |
> | :---: | :---: | :---: | :---: | :---: | :---: | :---: | :---: | :---: |
> |0.2 | 0.717 | 0.758 | 32.27 | 31.98 | 0.311 | 0.309 | 6.331 | 6.220|
> |0.4 | 0.684 | 0.692 | 32.06 | 31.07 | 0.309 | 0.309 | 6.612 | 5.992|
> |0.6 | 0.604 | 0.639 | 31.27 | 29.82 | 0.309 | 0.306 | 6.522 | 5.736|
> |0.8 | 0.416 | 0.661 | 33.02 | 29.29 | 0.311 | 0.306 | 7.339 | 5.562|
>
> ### **Timestep-wise Analysis of DC Lock-in**
> - We additionally conducted a timestep-wise analysis by partitioning the denoising trajectory into early, middle, and late windows (5 steps each). We found that cross-seed DC similarity shows a clear decreasing trend toward later timesteps. The DC energy ratio is also highest in the early regime, and the corresponding intervention causes the largest perceptual change. Consistent with the $\tau$ ablation in Fig. 6, these findings suggest that the lock-in effect is primarily an early-stage phenomenon. While increasing the $\tau$-window naturally increases the overall image shift, prolonging it into the later refinement stages eventually introduces unnecessary constraints on fine-grained synthesis, which can lead to a decline in visual fidelity. This supports using a relatively small 𝜏 in practice.
>
> | | Early (0-4) | Mid (12-16) | Late (24-27) |
> | :--- | :---: | :---: | :---: |
> | Cross-seed DC similarity | 0.975 | 0.893 | 0.756 |
> | DC energy ratio | 0.549 | 0.304 | 0.298 |
> | LPIPS | 0.717 | 0.113 | 0.095 |
>
> ### **Writing Improvement**
> - We thank the reviewer for their meticulous reading. All identified typos will be corrected, and the revised manuscript will be thoroughly proofread to ensure clarity and precision.

---

> > ### Author Rebuttal · Reviewer_i8dp · 2026-04-04
> >
> > Thanks you for your detailed responses and additional experiments. My concerns have been fully resolved and I will raise my rating to reflect it.

---

> > > ### Author Response · Authors · 2026-04-04
> > >
> > > We are deeply grateful for the time and effort you dedicated to reviewing our rebuttal and for your subsequent favorable feedback. Your insights have significantly strengthened the paper’s rigor, and we have ensured that all addressed points are fully reflected in the updated manuscript.

---

### Official Review · Reviewer_uGg4 · 2026-03-10

**Soundness:** 3
**Presentation:** 4
**Significance:** 3
**Originality:** 3
**Overall Recommendation:** 4
**Confidence:** 3

**Summary:**

This paper introduces DC Attenuation for diVersity Enhancement (DAVE), a training-free intervention designed to improve the sample diversity of flow-based text-to-image models. The authors observe that the spatial mean, or DC component, of intermediate Transformer features rapidly homogenizes across different noise seeds during the early stages of generation, creating a structural lock-in that limits output variation. To counteract this, DAVE selectively attenuates this low-frequency component within specific transformer blocks during the initial sampling steps. Extensive experiments on models like Stable Diffusion 3, Stable Diffusion 3.5, FLUX, and SANA1.5 demonstrate that this lightweight representational modulation enhances diversity metrics such as Recall and Vendi score while maintaining text alignment and image quality

**Compliance With Llm Reviewing Policy:**

Affirmed.

**Final Justification:**

The rebuttal strengthens the paper. I maintain my weak accept score.

**Key Questions For Authors:**

How sensitive is the identification of the target block pool to the choice of the prompt dataset used during the initial cross-seed cosine similarity analysis?

Since the analysis shows that manipulating different blocks yields specific attribute changes like size or color, is it possible to dynamically route the DC attenuation to different blocks during a single generation step to achieve a targeted composition of diversity?

How does DAVE perform under highly constrained or overly specific prompts where the desired structural diversity naturally approaches zero?

**Limitations:**

The authors adequately discuss the operational limits of their parameter selections, providing ablation studies on the temporal cutoff, attenuation strength, and target block pools.

**Strengths And Weaknesses:**

A major strength of this work is its mechanistic approach to diversity collapse, grounding the proposed solution in a clear analysis of early-stage spectral bias and the resulting trajectory lock-in. The intervention itself is highly efficient, requiring no retraining, no auxiliary optimization loops, and only negligible computational overhead compared to sampling-level heuristics like Particle Guidance. Furthermore, the empirical validation is comprehensive, spanning multiple state-of-the-art architectures and demonstrating a favorable trade-off between diversity and prompt faithfulness.

A notable weakness is the reliance on model-specific hyperparameters, such as the target block pool and the attenuation strength, which require empirical tuning or pre-computation for new architectures.

Additionally, the paper notes that specific blocks disproportionately affect distinct attributes like color or texture, suggesting that uniform DC attenuation might inadvertently cause entangled attribute shifts rather than purely structural diversity

---

> ### Author Rebuttal · Authors · 2026-03-30
>
> We thank the reviewer for the thoughtful questions regarding hyperparameter dependence, constrained prompts, and block-specific effects. To address these points, we have conducted additional experiments and will incorporate all the new results and analyses into the revised version of the paper to further strengthen our claims.
>
> ### **Sensitivity on hyperparameters ($\mathcal{L}$)**
> - **Unified $\mathcal{L}$ selection rule across architectures**: Regarding the concern about model-specific hyperparameters, we would like to clarify that DAVE employs a unified selection rule across architectures, where the target block pool $\mathcal{L}$ is defined by early cross-seed DC alignment. Since architectures vary in block structures and counts, the differences in selected indices naturally arise from applying this consistent criterion rather than per-model heuristics. While this criterion is broadly applicable, the selection process can be further streamlined by leveraging established module-specific insights [1, 2] as a principled starting point to identify key candidate blocks. Importantly, we observe that the DC-aligned blocks identified by our method are largely consistent with these established modules, further validating the reliability of our criterion. Consequently, this unified rule enables seamless adaptation to new architectures, requiring only a one-time calibration rather than exhaustive tuning.
> - **Robustness of $\mathcal{L}$ selection across prompts**: We further examined the dependence of the target block pool $\mathcal{L}$ on the calibration prompt set. To test this, we compared $\mathcal{L}$ pools independently constructed for 20 randomly sampled MSCOCO captions and 20 ImageNet labels. In SD3.5, these prompt-wise pools showed near-complete overlap (average pairwise Jaccard similarity = 0.98). A chi-square independence test across identified blocks also confirmed no significant association on the prompt source ($p=0.746$, Cramer's $V=0.154$). Furthermore, we conducted a cross-dataset evaluation by applying the $\mathcal{L}$ pool identified from one source to the other; the diversity gains remained robust, showing only a slight decrease relative to the in-dataset setting. Together, these results demonstrate the intrinsic robustness of $\mathcal{L}$ selection.
> | |Vendi (↑) | CLIP (↑) |
> | :--- | :---: | :---: |
> |Original | 1.4541 | 0.30883|
> | **DAVE** | 2.0275 | 0.30614|
> |Cross-dataset| 1.884 | 0.30611|
>
> - For $\alpha$ and $\tau$, we provide detailed sensitivity analysis in our response to *Reviewer i8dp*.
>
> ### **DAVE with Highly Constrained Prompts**
> - We evaluated DAVE on PartiPrompts [3], a benchmark categorized by varying complexity levels. Although complex prompts naturally allow less variation than simpler ones, DAVE consistently improves Vendi-score over the original model across all prompt complexity levels. Notably, CLIP-score changes remain minimal (absolute gap < 0.01), demonstrating that DAVE enhances meaningful diversity without compromising prompt alignment, even under high structural constraints.
>
> - Vendi Score
> | Method | Basic | Simple Detail | Fine-grained | Complex |
> | :--- | :---: | :---: | :---: | :---: |
> | Original | 2.0513 | 1.5754 | 1.5244 | 1.5107 |
> | **DAVE** | 2.4655 | 2.0233 | 2.0087 | 2.0060 |
>
> - CLIP Score
> | Method | Basic | Simple Detail | Fine-grained | Complex |
> | :--- | :---: | :---: | :---: | :---: |
> | Original | 0.2883 | 0.3443 | 0.3313 | 0.3207 |
> | **DAVE** | 0.2978 | 0.3368 | 0.3288 | 0.3111 |
>
> ### **Block-specific roles and dynamic routing.**
> - To address the potential for dynamic routing, we evaluated manipulating multiple blocks concurrently during a single generation step. While a targeted composition of diversity is possible through such multi-block attenuation, we found that distributing the attenuation across blocks tends to weaken individual attribute control (VLM-based property score: color 0.8→0.5, size 0.9→0.65). These results suggest that while this is a promising direction, a more careful treatment would require deeper analysis of block-wise and step-wise routing strategies, which we leave for future work.
>
> ---
>
> *Reference*
> [1] Li, Binglei, et al. "Unraveling MMDiT Blocks: Training-free Analysis and Enhancement of Text-conditioned Diffusion." (2026).
> [2] Yang, Yitong, et al. "SplitFlux: Learning to Decouple Content and Style from a Single Image." (2025).
> [3] Yu, Jiahui, et al. "Scaling autoregressive models for content-rich text-to-image generation." (2022).

---

> > ### Author Rebuttal · Reviewer_uGg4 · 2026-04-04
> >
> > The rebuttal strengthens the paper with targeted evidence on prompt-set robustness of block-pool selection, behavior under highly constrained prompts, and the limits of multi-block routing. These additions make the method look less ad hoc than in the original submission and support the central claim that early DC attenuation produces a genuine diversity gain across settings.
> >
> > I remain positive because the core idea is novel, lightweight, and practically useful, but I still see this as a solid weak accept rather than a clear accept.

---

> > > ### Author Response · Authors · 2026-04-04
> > >
> > > We sincerely thank you for the thorough assessment of our rebuttal and for your favorable feedback. We are encouraged that the additional evidence helped clarify the systematic nature of our approach and its core benefits. All insights will be fully integrated to ensure the highest level of rigor in the final manuscript.

---

### Official Review · Reviewer_1Fmt · 2026-03-12

**Soundness:** 3
**Presentation:** 4
**Significance:** 3
**Originality:** 4
**Overall Recommendation:** 5
**Confidence:** 3

**Summary:**

This paper studies the problem of low diversity in text-to-image generation. The authors analyze the DC component of transformer representations and show that, during early sampling steps, this component remains consistent across different seeds, thereby constraining the generative trajectory. Based on this observation, the paper proposes DAVE,  a method that attenuates the DC component during inference. Authors evaluate DAVE on several text-to-image models and report improved diversity, while maintaining competitive image quality and text alignment, without requiring additional training.

**Compliance With Llm Reviewing Policy:**

Affirmed.

**Final Justification:**

The new results address my main concerns. In particular, the extended baseline comparison makes the evaluation much more convincing. Based on these results, I now see that DAVE performs well not only against older baselines, but also against more recent diversity methods.

The additional experiments on distilled models and with CFG are also important. It is good to see that DAVE remains effective and outperforms CADS and PG in the distilled setting. The CFG results also clarify that DAVE is compatible with CFG and remains useful at higher guidance scales, which was a major missing point in the original submission.

Based on the additional results, I am increasing my overall recommendation to 5: Accept.

**Key Questions For Authors:**

Questions

1. Why were more recent diversity methods, such as Shielded Guidance and DiverseFlow, not included in the comparison? Since these methods are mentioned in the paper, it would be helpful to understand whether they were considered and, if so, why they were excluded.

2. Why are CADS and PG not evaluated on the distilled models in Table 3? If these methods do not apply in those settings, please state this clearly.

3. Did you evaluate DAVE together with CFG? Since CFG is a standard component in text-to-image generation, it is important to understand whether DAVE is intended to work together with CFG or to replace it.

**Limitations:**

Yes

**Strengths And Weaknesses:**

Paper Strengths

1. The paper is well written and easy to follow. The main claims are supported by empirical results and theoretical analysis.

2. The proposed method is simple and computationally efficient.

3. The paper studies the diversity problem from the perspective of transformer representations, which is an interesting direction.

4. The paper includes extensive ablations on hyperparameters.

Major Weaknesses

1. The baseline set seems incomplete. The paper compares against older methods and does not include newer methods that the authors mention themselves (Shielded Guidance, DiverseFlow).

2. The paper does not evaluate DAVE with CFG. This is a very important omission, since CFG is a standard component in text-to-image generation. For example, CADS was evaluated using CFG with high guidance scales.

Minor Weaknesses

1. On distilled models (Table 3), CADS and PG are not evaluated.

2. The abstract appears to contain a typo: "infivsyinh".

---

> ### Author Rebuttal · Authors · 2026-03-30
>
> We sincerely appreciate the reviewer for the constructive feedback.In the revised paper, we will incorporate additional experimental results to further validate our findings and provide a more comprehensive evaluation.
>
> ### **Performance Comparison against Extended Baselines**
> - To address the concern about an incomplete baseline set, we substantially expanded the comparison on Stable Diffusion 3.5 by adding SPELL [1], Diverseflow [2], Sparke [3], and OSCAR [4]. To maintain comparative integrity, we implemented SPELL and DiverseFlow strictly according to their original pseudocode and default settings, as no public codebases were available. Despite SPARKE’s slightly higher Vendi score, DAVE demonstrates comprehensive superiority by achieving substantially stronger Recall with comparable CLIP. Qualitative results in Fig. 12 further illustrate that the additional diversity is not driven by artifacts. As a practical note, most of these baselines rely on reference-set information and therefore require nontrivial batch size or memory, whereas DAVE does not.
> | Method | FID ($\downarrow$) | Precision ($\uparrow$) | Recall ($\uparrow$) | Vendi ($\uparrow$) | CLIP ($\uparrow$) |
> | :--- | :--- | :--- | :--- | :--- |:--- |
> |Original | 36.37 | 0.821 | 0.255| 0.293 | 0.313 |
> | SPELL  | 37.13 | 0.815 | 0.235 | 1.699| **0.318** |
> | DiverseFlow | 39.78 | 0.818| 0.209 | 1.557 | 0.313|
> |Sparke | 35.39 | 0.792| 0.282| **2.468** | 0.311 |
> |Oscar | 35.87 | **0.827** | 0.245| 0. 169| 0.312 |
> |**DAVE**| **29.75** | 0.730 | **0.441** | 2.370 | 0.308|
>
> ### **Comparison within Distilled Model Settings**
> - We also added CADS and PG as additional baselines for comparison in the distilled-model setting in Section 5.2 (Stable Diffusion 3.5 Large Turbo). DAVE remains effective in this few-step regime and still provides the strongest diversity gain with only marginal changes in fidelity and alignment.
> | Method | Precision ($\uparrow$) | Recall ($\uparrow$) | Vendi ($\uparrow$) | CLIP ($\uparrow$) |
> | :--- | :---: | :---: | :---: | :---: |
> |Original | **0.816** | 0.384 | 1.393 | 0.205 |
> | CADS | 0.785 | 0.423 | 1.565 | 0.206 |
> | PG | 0.788 | 0.418 | 1.442 | **0.211** |
> | **DAVE** | 0.798 | **0.431** | **1.645** | 0.205 |
>
> ### **Compatibility and Synergy with Classifier-Free Guidance (CFG)**
> - DAVE is compatible with CFG and can be naturally combined with it to further enhance performance. Across all CFG scales, DAVE substantially improves Recall, and at moderate-to-high CFG scales it also improves Vendi, while keeping CLIP nearly unchanged. This is practically important because higher CFG typically improves fidelity at the cost of diversity, whereas DAVE helps recover diversity under such settings without harming text alignment. Note that our main experiments already follow the default CFG setting of each base model, showing that DAVE works under standard practical configurations.
> | CFG | Method | FID ($\downarrow$) | Precision ($\uparrow$) | Recall ($\uparrow$) | CLIP ($\uparrow$) | Vendi ($\uparrow$) |
> | :--- | :---: | :---: | :---: | :---: | :---: |:---: |
> |3 | Original | 127.59 | 0.860 | 0.0524 | 0.313 | 2.540 |
>  | | DAVE | 100.80 | 0.752 | 0.165 | 0.310 | 2.307 |
> |5 | Original | 138.81 | 0.883 | 0.016 | 0.314 | 2.975 |
> | | DAVE | 108.29 | 0.795 | 0.133 | 0.312 | 2.995 |
> |7 | Original | 138.82 | 0.871 | 0.015 | 0.316 | 3.385 |
> | | DAVE | 108.65 | 0.735 | 0.132 | 0.313 | 3.828 |
> |10 | Original | 132.27 | 0.854 | 0.035 | 0.318 | 4.189 |
> | | DAVE | 104.86 | 0.634 | 0.162 | 0.307 | 5.370 |
>
> ### **Correction of Typographical Errors**
> - We apologize for the typo in the abstract. The term will be corrected to "suggesting" in the final version. Additionally, we are re-examining the entire manuscript to improve the overall writing quality and technical accuracy of the paper.
> ---
> *Reference*
> [1] Kirchhof, Michael, et al. "Shielded Diffusion: Generating Novel and Diverse Images using Sparse Repellency." (2024).
> [2] Morshed, Mashrur M., and Vishnu Boddeti. "Diverseflow: Sample-efficient diverse mode coverage in flows." (2025).
> [3] Jalali, Mohammad, et al. "Sparke: Scalable prompt-aware diversity guidance in diffusion models via rke score." (2025).
> [4] Wu, Jingxuan, et al. "OSCAR: Orthogonal Stochastic Control for Alignment-Respecting Diversity in Flow Matching." (2025).

---

> > ### Author Rebuttal · Reviewer_1Fmt · 2026-04-02
> >
> > I appreciate the detailed response and the additional experiments.
> >
> > The new results address my main concerns. In particular, the extended baseline comparison makes the evaluation much more convincing. Based on these results, I now see that DAVE performs well not only against older baselines, but also against more recent diversity methods.
> >
> > The additional experiments on distilled models and with CFG are also important. It is good to see that DAVE remains effective and outperforms CADS and PG in the distilled setting. The CFG results also clarify that DAVE is compatible with CFG and remains useful at higher guidance scales, which was a major missing point in the original submission.
> >
> > Based on the additional results, I am increasing my overall recommendation to 5: Accept.

---

> > > ### Author Response · Authors · 2026-04-02
> > >
> > > Thank you for your positive feedback and for the score update. We deeply value your constructive insights, which have been instrumental in refining our work. All experimental results and discussions addressing your points have been incorporated into the revised manuscript to ensure a more rigorous presentation of our findings.

---

### Decision · Program_Chairs · 2026-04-30

**Decision:**

Accept (regular)

**Comment:**

This paper was reviewed by 3 knowledgeable reviewers. The reviewers raised the following main concerns:
1. Incomplete baseline comparisons (e.g. Shielded Guidance/SPELL, DiverseFlow, SPARKE, OSCAR, CADS and PG on distilled models) and CFG comparisons (1Fmt).
2. Model-specific hyper-parameter tuning required (uGg4, i8dp) and unclear hyper-parameter sensitivity (i8dp).
3. Scale-preserving variant not explored (i8dp).
4. Timestep-wise DC behavior (i8dp).
5. Different blocks affect different attributes, uniform DC attenuation may cause unintended entangled changes (uGg4).
6. Unclear behavior under highly specific prompts (uGg4).
7. Presentation quality could be improved (multiple typos) (i8dp, 1Fmt).

The rebuttal addressed the reviewers concerns by adding comparisons against requested baselines and increasing CFG values, showing that the proposed approach still provided compelling advantages. The rebuttal clarified that a unified selection rule is applied across architectures, showed that cross-dataset transfer only shows slight diversity decrease, and introduced ablation to study hyper-parameter sensitivity. The rebuttal also the considered the scale-preserving variant mentioned by the reviewers, conducted a timestep analysis partitioning the denoising trajectory into early/middle/late windows, tested multi-block concurrent attenuation for dynamic routing, and presented  results on PartiPrompts across complexity levels.

After rebuttal, all reviewers' concerns are resolved, the extended baselines and experiments make the evaluation of the proposed approach compelling, and reviewers unanimously recommend acceptance.